# Sinkhorn Normalization of Diffusion Kernels

Nathan Kessler [* 1 2]   Robin Magnet [* 2]   Jean Feydy [2]

## Abstract

Smoothing a signal based on local neighborhoods is a core operation in machine learning and geometry processing. On well-structured domains such as vector spaces and manifolds, the Laplace operator derived from differential geometry offers a principled approach to smoothing via heat diffusion, with strong theoretical guarantees. However, constructing such Laplacians requires a carefully defined domain structure, which is not always available. Most practitioners thus rely on simple convolution kernels and message-passing layers, which are biased against the boundaries of the domain. We bridge this gap by introducing a broad class of *smoothing operators*, derived from general similarity or adjacency matrices, and demonstrate that they can be normalized into *diffusion-like operators* that inherit desirable properties from Laplacians. Our approach relies on a symmetric variant of the Sinkhorn algorithm, which rescales positive smoothing operators to match the structural behavior of heat diffusion. This construction enables Laplacian-like smoothing and processing of irregular data such as point clouds, sparse voxel grids or mixture of Gaussians. We show that the resulting operators not only approximate heat diffusion but also retain spectral information from the Laplacian itself, with applications to shape analysis and matching. Code is available at github.com/RobinMagnet/SinkhornKernels

## 1. Introduction

**Discrete Differential Geometry.** Geometric data analysis provides a principled framework for understanding complex data (Gallot et al., 2004; Bronstein et al., 2021).

*Equal contribution [1]Centre Borelli, ENS Paris-Saclay, France [2]Inria, Université Paris Cité, Inserm, HeKA, F-75015 Paris, France. Correspondence to: Robin Magnet <robin.magnet@inria.fr>, Nathan Kessler <nathan.kessler@ens-paris-saclay.fr>.

*Proceedings of the 43rd International Conference on Machine Learning*, Seoul, South Korea. PMLR 306, 2026. Copyright 2026 by the author(s).

These tools work particularly well in two or three dimensions, though they naturally extend to graphs and higher-dimensional domains. While differential geometry provides elegant constructions on well-structured data like triangle meshes (Crane, 2018; Botsch et al., 2010), adapting these tools to less structured representations like point clouds or voxel grids remains a major challenge (Barill et al., 2018; Lachaud et al., 2023; Feng & Crane, 2024).

High-quality triangular meshes are often unavailable in practice, due to the acquisition process (Bogo et al., 2017) or the nature of the data itself (Marcus et al., 2007). For example, medical imaging relies on voxelized segmentation masks (Marcus et al., 2007), and anatomical structures such as trabecular bones cannot be easily described as surfaces. Point clouds and recent representations like Gaussian splats (Kerbl et al., 2023; Zhou & Lähner, 2025) are now popular, but lack explicit connectivity. Applying classical geometric operators here usually require local approximations of the underlying manifold (Sharp & Crane, 2020; Belkin et al., 2009; Zhou & Lähner, 2025), which introduce errors and degrade performance.

**Smoothing Operations.** Transferring geometry-aware methods designed for clean meshes to general unstructured data representations is essential for scalable learning with shapes. Among the most fundamental operations in geometry processing is *smoothing*, which modifies a signal using local geometric information (Taubin, 1995; Sharp et al., 2022). A canonical example is *heat diffusion*, controlled on meshes by the matrix exponential of the Laplacian (Gallot et al., 2004). This process is central to many pipelines for shape analysis (Crane et al., 2017), segmentation (Sharp et al., 2022), and correspondence (Sun et al., 2009).

**Contributions.** In this work, we develop stable smoothing operators acting at a *fixed scale* on unstructured discrete data. To do so, we generalize heat diffusion to arbitrary discrete domains, including point clouds, voxel grids, Gaussian mixtures and binary masks. Given any symmetric similarity matrix, our method produces a heat-diffusion-like operator via *symmetric Sinkhorn normalization* (Knight et al., 2014). The resulting linear operator is symmetric with respect to a mass-weighted inner product, and its spectrum approximates the exponential of a Laplacian, as illustrated in Figure 1. This approach notably guarantees *mass conser-*

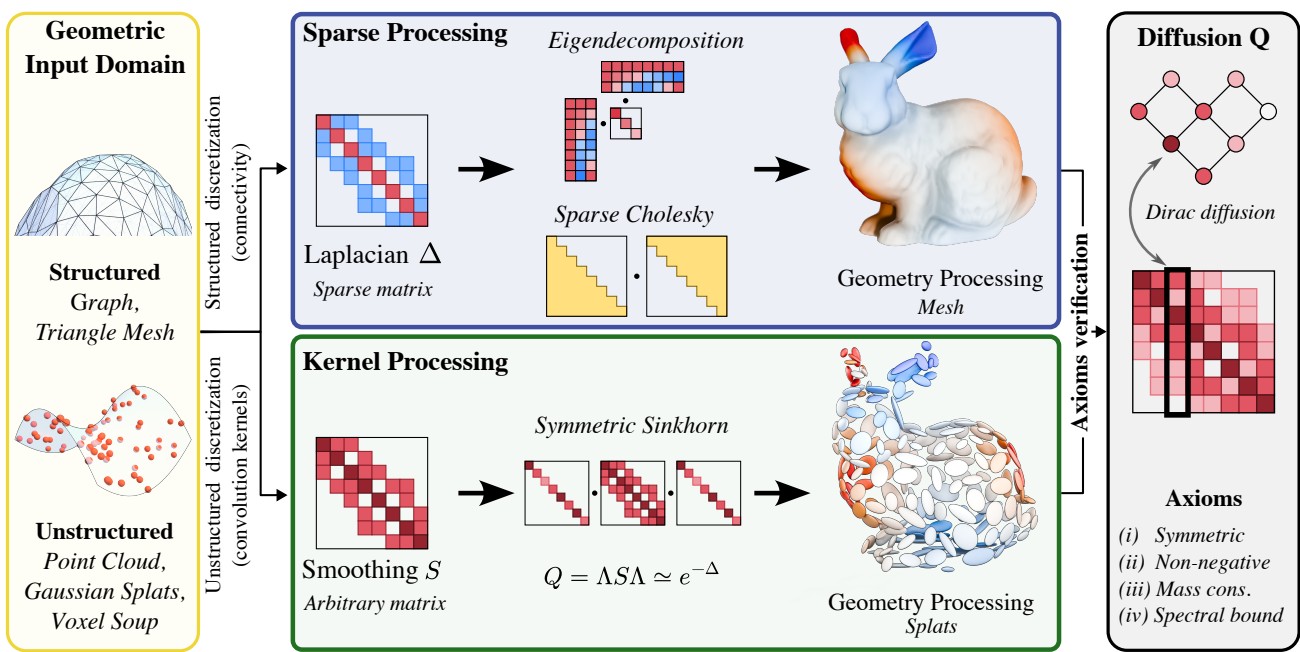

*Figure 1.* Turning smoothing operators into heat-like diffusion operators. Heat diffusion is a fundamental operation on geometric domains, but standard approaches only apply to *structured inputs* that provide explicit connectivity, used to assemble a sparse Laplacian $\Delta$. Heat diffusion is then obtained using sparse factorization or truncated eigendecomposition. *Unstructured* representations, such as Gaussian splats or point clouds have no natural Laplacian, but admit cheap smoothing (or convolutions) kernels. We show that an *arbitrary* smoothing operator $S$ can be corrected into a diffusion operator $Q$ via symmetric Sinkhorn normalization, without any connectivity or factorization. The resulting operator shares key practical axioms of diffusion, notably mass preservation of the signal.

*vation*, meaning that the total sum of the signal if preserved under diffusion, corresponding to physical conservation of heat over the domain.

Our contributions can be summarized as follows:

- We present an algebraic framework for defining smoothing, Laplacians, and diffusions on general discrete geometric representations, clarifying their shared structure and assumptions.
- We propose an efficient and GPU-friendly algorithm to transform *arbitrary* similarity matrices into mass-preserving diffusion operators. While inspired by recent theoretical works in manifold learning (Wormell & Reich, 2021; Cheng & Landa, 2024), we focus on stability at *fixed diffusion scale*, instead of asymptotic limits.
- We demonstrate broad applicability to geometric data analysis, from spectral shape analysis and generative modelling to state of the art shape correspondence on unstructured data.

## 2. Related Works

**Laplacians and Heat Diffusions.** The Laplace–Beltrami operator $\Delta$ is essential in discrete geometry processing on triangle meshes (Sorkine, 2005; Botsch et al., 2010), with spectral properties used for shape analysis (Reuter et al., 2006) and correspondence (Levy, 2006; Ovsjanikov et al., 2012). A closely related tool is the heat equation $\partial_t f = -\Delta f$, whose solution $f(t) = e^{-t\Delta} f_0$ smooths initial signals $f_0$. Heat diffusion enables computing shape descriptors (Sun et al., 2009; Bronstein & Kokkinos, 2010), geodesic distances (Crane et al., 2017; Feng & Crane, 2024) parallel transport (Sharp et al., 2019b) or shape correspondences (Vestner et al., 2017; Cao et al., 2025). Heat smoothing has also inspired neural architectures for geometric learning (Sharp et al., 2022; Gao et al., 2024) and generative models (Yang et al., 2023). In practice, $e^{-t\Delta}$ is computed via implicit Euler integration (Botsch et al., 2010) or spectral truncation (Sharp et al., 2022), relying on mesh discretizations (Pinkall & Polthier, 1993; Meyer et al., 2003; Sharp et al., 2019a) and pre-computed factorizations. Extensions to non-manifold triangulations have been proposed (Sharp & Crane, 2020; Belkin et al., 2009), but generalizing to high-dimensional or unstructured data such as noisy point clouds and sparse voxel grids remains challenging.

**Graph Laplacians.** The Laplacian also plays a central role in graph-based learning and analysis (Chung, 1997), supporting spectral clustering (von Luxburg, 2007), embedding, and diffusion. Unlike mesh cotangent Laplacian (Pinkall & Polthier, 1993; Meyer et al., 2003), graph Laplacians relies only on connectivity and edge weights,

enabling broader applicability. Discrete approximations of heat diffusion, such as the explicit Euler step $I - t\Delta$, form the basis of many graph neural networks (Kipf & Welling, 2017; Hamilton et al., 2017; Chamberlain et al., 2021), with recent works exploring approximate implicit schemes (Behmanesh et al., 2023; Chamberlain et al., 2021), which however lose key properties or limit the flexibility of the diffusion. Beyond Laplacians, message passing (Gilmer et al., 2017) or graph attention (Veličković et al., 2018) offer alternative forms of smoothing. In this work, we propose *general axioms* for smoothing operators on discrete domains, formalizing desirable properties such as symmetry or mass conservation. We show that popular approaches, as well as various Laplacian normalization techniques (symmetric, random-walk) can be interpreted as specific cases in this framework. These classical methods often trade-off symmetry or mass preservation, while we leverage *Sinkhorn normalization* that satisfies both. This produces operators that are symmetric under a mass-weighted inner product and spectrally similar to Laplacian exponentials, while remaining compatible with unstructured geometric data.

**Sinkhorn Scaling.** The Sinkhorn algorithm (Sinkhorn & Knopp, 1967) scales a nonnegative matrix into a doubly stochastic one via alternating row and column normalizations, and is widely used in machine learning (Cuturi, 2013) thanks to its efficiency and GPU compatibility. For symmetric inputs, a symmetry-preserving variant exits (Knight et al., 2014). Recent work in manifold learning related to diffusion maps (Coifman & Lafon, 2006) applies Sinkhorn normalization to graph Laplacians and Gaussian kernels on point clouds (Marshall & Coifman, 2019; Wormell & Reich, 2021; Cheng & Landa, 2024). These works focus on the asymptotic limit to an underlying smooth Laplace operator when the scale $\sigma \to 0$ and sampling density increases. In contrast, we are interested in geometry processing and in *the smoothing operation itself*, at fixed scale. To this end, we obtain stability results for fixed bandwidth and increasing density, and develop an axiomatic framework which corrects *any smoothing operator* so that it behaves similarly to heat diffusion. In particular, we ensure exact mass preservation and symmetry under a mass-weighted inner product.

**Generalization to Unstructured Data.** Extending differential operators to unstructured data remains an open challenge. Irregular representations, such as point clouds or gaussian splats, typically rely on local approximations or learned kernels (Wu et al., 2019; Sharp & Crane, 2020; Sharp et al., 2022; Zhou & Lähner, 2025). Our framework requires only similarity between points, providing a unified approach to smoothing across these modalities, without consistent manifold assumptions or expensive linear algebra routines.

**Motivation and Contribution.** Heat diffusion provides a natural smoothing operation on continuous manifolds, but its discretization is often expensive or compromises important properties such as mass preservation. On general discrete geometric domains, where Laplacians are not easily accessible, general smoothing kernels are often preferred in practice. However these un-normalized kernels usually lack some essential structural properties required for robust geometry processing, such as stability under varying sampling density or mass preservation. In this work, we show that such arbitrary smoothing operators can be corrected at minimal computational cost to mimic the key properties of heat diffusion, without any knowledge of an underlying Laplacian. To this end, Section 3 provides intuition and a basic construction on simple unweighted graphs, while Section 4 describes the full theoretical framework for our diffusion operators.

## 3. Warm-up: Graphs

To build intuition about our framework, we begin with a simple example on an unweighted undirected graph $\mathcal{G}$ with vertex set $\mathcal{V}$ and edge set $\mathcal{E}$ (see Figure 2). In this section, we illustrate two common approaches to smoothing a signal $f \in \mathbb{R}^{\mathcal{V}}$ defined on the graph vertices.

**Laplacian Smoothing.** A common method to quantify the regularity of $f$ is to use a discrete derivative operator $\delta \in \{0, \pm 1\}^{\mathcal{E} \times \mathcal{V}}$, which encodes differences across edges. This induces a Dirichlet energy $E(f) = \frac{1}{2}\|\delta f\|^2 = \frac{1}{2}(\delta f)^\top \delta f$, that can be rewritten as $E(f) = \frac{1}{2}f^\top \Delta f$, where $\Delta = \delta^\top \delta$ is a symmetric positive semi-definite matrix acting as a discrete Laplacian. The matrix $\Delta$ admits an orthonormal eigen-decomposition with eigenvectors $(\Phi_i)$ and non-negative eigenvalues $(\lambda_i)$. Low values of $E(\Phi_i) = \lambda_i/2$ correspond to smoother eigenfunctions. These eigenvectors can be interpreted as frequency modes: $\lambda_1 = 0$ corresponds to the constant function, while higher eigenvalues $\lambda_i$ capture finer variations.

Heat diffusion, governed by the operator $\mathcal{H}_t = e^{-t\Delta}$, acts as a smoothing transform that damps high-frequency components while preserving the low-frequencies. If $f = \sum_i \widehat{f}_i \Phi_i$, then:

$$E(\mathcal{H}_t f) = \frac{1}{2}\sum_i \lambda_i e^{-2t\lambda_i} \widehat{f}_i^2 \leq E(f). \qquad (1)$$

By construction, diffusion regularizes input functions and converges as $t \to \infty$ to a constant signal $\widehat{f}_1$. Remarkably, the *total mass* of the signal is also preserved for all $t$: if $\langle \cdot, \cdot \rangle$ denotes the dot product and $\mathbf{1}$ is the constant vector, then $\langle \mathbf{1}, \mathcal{H}_t f \rangle = \langle \mathbf{1}, f \rangle$. This follows from the symmetry of $e^{-t\Delta}$ and the fact that constant functions are fixed points of the heat flow: $\mathcal{H}_t \mathbf{1} = \mathbf{1}$. Lastly, since $-\Delta$ is a Metzler matrix

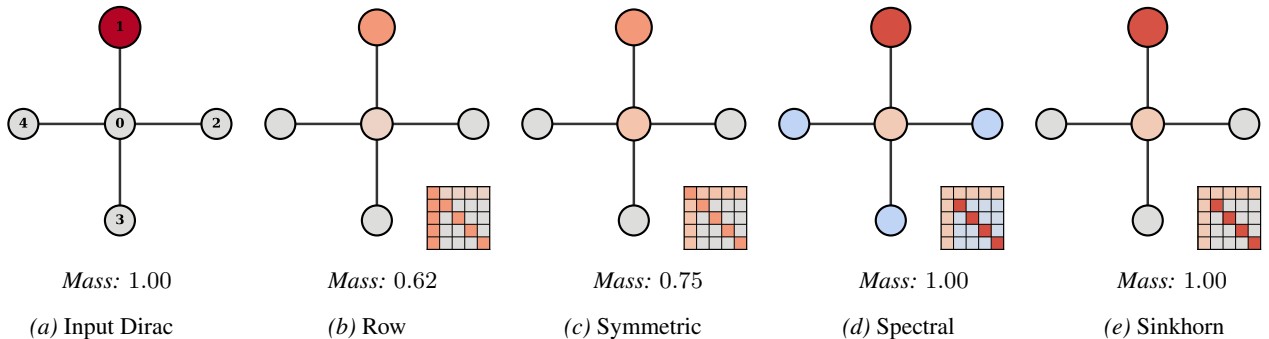

*Figure 2.* Diffusion of a Dirac function for different normalizations of the raw smoothing operator $K$. Row and symmetric normalizations distort mass, while a truncated spectral approximation that uses 4 out of 5 eigenvectors introduces negative values (in blue). In contrast, our symmetric Sinkhorn normalization preserves both positivity and mass. The full diffusion matrices are shown as insets

(non-negative off-diagonals), $\mathcal{H}_t$ is entrywise non-negative, preserving signal positivity (Berman & Plemmons, 1994).

While the Laplacian exponential provides strong regularization properties, its computation is expensive in practice. Most authors rely on the implicit Euler scheme $(I + t\Delta)^{-1} f$ which guarantees numerical stability but requires solving a linear system for every new signal $f$.

**Message-passing and Local Averages.** A common alternative is to apply local averaging operators $K$ derived from the graph's adjacency matrix $A$. One possible choice is to use the linear operator $K = \frac{1}{2}(D + A)$, where $D = \text{diag}(A1)$ is the diagonal degree matrix. However, this raw smoothing operator $K$ lacks key properties of $e^{-t\Delta}$ and is highly sensitive to vertex degrees: well-connected vertices disproportionately influence the output, while low-degree nodes contribute less. This introduces a bias at boundaries of the domain that is undesirable in many applications.

Several methods have been proposed to mitigate this issue (Chung, 1997; Coifman & Lafon, 2006):

1. **Row normalization** divides each row of $K$ by the degree of the corresponding vertex. This yields the operator $D^{-1}K$ that performs a local averaging but breaks symmetry.
2. **Symmetric normalization** restores symmetry via $D^{-\frac{1}{2}} K D^{-\frac{1}{2}}$, but does not preserve constant signals.
3. **Spectral methods** approximate the heat diffusion operator $e^{-t\Delta}$ as a low-rank matrix $\sum_{i=1}^{R} e^{-t\lambda_i} \Phi_i \Phi_i^\top$ in the span of the first $R$ eigenvectors of the Laplacian $\Delta$. While this preserves desirable diffusion properties, computing these eigenvectors is expensive and truncation introduces undesirable ringing artifacts.

**Issues and Normalization.** As illustrated in Figures 2b to 2d, existing normalization strategies fail to fully capture the desired properties of heat diffusion. The *Sinkhorn* or *bi-stochastic* scaling algorithm offers a principled and fast

alternative, that combines the benefits of both row-wise and symmetric normalization, while avoiding the artifacts of spectral methods. The core idea is to apply symmetric normalization iteratively until convergence. Under mild assumptions, developed in Section 4, there exists a unique positive diagonal matrix $\Lambda$ such that the rescaled operator $\Lambda K \Lambda$ is both symmetric and mass-preserving, as illustrated in Figure 2e.

Related work on manifold learning has studied Sinkhorn normalization for graph Laplacian derived from points clouds in $\mathbb{R}^d$ (Marshall & Coifman, 2019; Wormell & Reich, 2021; Cheng & Landa, 2024), using Gaussian kernels $K$, and with a specific focus on the joint limit of increasing sample density and vanishing bandwidth. We are instead interested in correcting an *arbitrary* averaging operator on any discrete geometric domain, for which the following section provides the full theoretical framework.

## 4. Theoretical analysis

**Notations.** Let $\mathcal{X} \subset \mathbb{R}^d$ be a bounded domain with a positive Radon measure $\mu$. We consider signals $f : \mathcal{X} \to \mathbb{R}$ in $L^2_\mu(\mathcal{X})$ with inner product $\langle f, g \rangle_\mu = \int_{\mathcal{X}} f(x)g(x)\, \mathrm{d}\mu(x)$. We define the *mass* of a function as $\langle f, 1 \rangle_\mu = \int_X f(x)\, \mathrm{d}\mu(x)$. For discrete $\mu = \sum_{i=1}^{N} m_i \delta_{x_i}$, functions $f$ become vectors $(f(x_1), \ldots, f(x_N))$ and $\mu$ corresponds to a positive diagonal matrix $M = \text{diag}(m_1, \ldots, m_N) \in \mathbb{R}^{N \times N}$. The inner product becomes $\langle f, g \rangle_\mu = f^\top M g$: we use $\langle \cdot, \cdot \rangle_\mu$ or $\langle \cdot, \cdot \rangle_M$ interchangeably.

We denote by $A^{\top_\mu}$ (or $A^{\top_M}$), the adjoint of $A : L^2_\mu(\mathcal{X}) \to L^2_\mu(\mathcal{X})$ with respect to the $\langle \cdot, \cdot \rangle_\mu$. In the finite case, $A^{\top_M} = M^{-1} A^\top M$, satisfying $\langle f, Ag \rangle_M = \langle A^{\top_M} f, g \rangle_M$, where $A^\top$ is the standard transpose. A matrix is symmetric with respect to $\langle \cdot, \cdot \rangle_M$ iff $S = KM$ with $K^\top = K$.

**Laplace-like Operators.** As introduced in Section 3, Laplacians capture local function variations. We highlight

the key structural properties of Laplacians, simplifying the list identified in Wardetzky et al. (2008) for meshes:

**Definition 4.1** (Laplace-like Operators). A *Laplace-like operator* is a linear map $\Delta : L_\mu^2(\mathcal{X}) \to L_\mu^2(\mathcal{X})$, identified with a matrix $(\Delta_{ij})$ in the discrete case, that satisfies the following properties:

$$
\begin{array}{ll}
(i)\ \Delta^{\top_\mu} = \Delta & (iii)\ \langle f, \Delta f \rangle_\mu \geq 0, \quad \forall f \in L_\mu^2(\mathcal{X}) \\
(ii)\ \Delta \mathbf{1} = 0 & (iv)\ \Delta_{ij} \leq 0, \quad \forall i \neq j
\end{array}
$$

Properties $(i)$–$(iii)$ reflect the classical self-adjoint positive semi-definite structure, typically obtained from integration by parts: $\langle \nabla f, \nabla g \rangle_\mu = \langle f, \Delta g \rangle_\mu$. Condition $(iv)$, which corresponds to a *Metzler* structure (Berman & Plemmons, 1994) in the discrete setting (*i.e.* non-negative off-diagonal entries), ensures intuitive diffusion behavior: diffusing a signal with $\partial_t f = -\Delta f$ causes mass to flow outwards (Wardetzky et al., 2008). We refer to Section A for a definition in the continuous case using Kato's inequality (Arendt, 1984).

These conditions hold for graph Laplacians. On triangle meshes, the cotangent Laplacian satisfies properties $(i)$–$(iii)$ by construction, but $(iv)$ only when angles are not obtuse. Violations of this condition, which results in undesirable positive weights, are a well-known issue in geometry processing, usually solved using intrinsic triangulations (Bobenko & Springborn, 2007; Sharp et al., 2019a).

**Diffusion Operators.** As presented in Section 3, the family of diffusion operators $e^{-t\Delta}$ associated to a Laplacian $\Delta$ play a central role in geometry processing and learning. These operators smooth input signals while preserving key structural properties. From the properties of Theorem 4.1 we define diffusion operators as follows:

**Definition 4.2** (Diffusion Operators). A *diffusion operator* is a linear map $Q : L_\mu^2(\mathcal{X}) \to L_\mu^2(\mathcal{X})$ that satisfies the following properties, where $\sigma(Q)$ is the spectrum of $Q$:

$$
\begin{array}{ll}
(i)\ Q^{\top_\mu} = Q & (iii)\ \sigma(Q) \subseteq [0,1] \\
(ii)\ Q\mathbf{1} = \mathbf{1} & (iv)\ f \geq 0 \implies Qf \geq 0
\end{array}
$$

Laplace-like operators and diffusion operators are almost equivalent: the exponential of an any Laplace-like operator is a diffusion operator, while the principal logarithm of a diffusion operator yields properties $(i)$–$(iii)$ of Theorem 4.1. Property $(iv)$ in Theorem 4.2 is slightly weaker: true equivalence would require $Q^t$ to be entrywise positive for all $t \geq 0$ (Section A).

These properties reflect the structure of classical heat diffusion. Symmetry and constant preservation imply *mass conservation*, since for any $f$, $\langle \mathbf{1}, Qf \rangle_\mu = \langle Q^{\top_\mu}\mathbf{1}, f \rangle_\mu = \langle Q\mathbf{1}, f \rangle_\mu = \langle \mathbf{1}, f \rangle_\mu$. Entrywise positivity $(iv)$ follows from the non-negativity of the heat kernel, with more details provided in Section A. Damping $(iii)$ ensures that repeated applications of $Q$ attenuate high-frequency components.

---

**Algorithm 1** Symmetric Sinkhorn Normalization

1: **Input:** Smoothing matrix $S \in \mathbb{R}^{N \times N}$    % ($S = KM$).
2: Initialize $\Lambda \leftarrow I_N$      % $\Lambda$ *is a diagonal matrix.*
3: **while** $\sum_i |\Lambda_{ii} \sum_j S_{ij}\Lambda_{jj} - 1| >$ tolerance **do**
4:     $d_i \leftarrow \sum_j S_{ij}\Lambda_{jj}$      % *Matvec product with* $S$.
5:     $\Lambda_{ii} \leftarrow \sqrt{\Lambda_{ii}/d_i}$      % *Coordinate-wise update.*
6: **end while**
7: **Return** $Q = \Lambda S \Lambda$      % $Q$ *is a positive scaling of* $S$.

---

Finally, when $Q = e^{-t\Delta}$, its leading eigenvectors are with the lowest-frequency modes of $\Delta$, with $\lambda_i^Q = e^{-t\lambda_i^\Delta}$. This allows to recover low-frequency Laplacian structure via power iterations on $Q$, without computing small eigenpairs directly (see Section 6).

**Smoothing Operators.** In practice, defining a diffusion operator without access to an underlying Laplacian can be challenging. Instead, many operators commonly used in geometry processing, such as adjacency or similarity matrices, implicitly encode local neighborhood structures and enable function smoothing through local averaging. We refer to such matrices as *smoothing operators*:

**Definition 4.3** (Smoothing Operators). A *smoothing operator* is a linear map $S : L_\mu^2(\mathcal{X}) \to L_\mu^2(\mathcal{X})$, identified with a matrix $(S_{ij})$ in the discrete case, that satisfies the following properties:

$$
\begin{array}{ll}
(i) & S^{\top_\mu} = S \\
(ii) & \langle f, Sf \rangle_\mu \geq 0, \quad \forall f \in L_\mu^2(\mathcal{X}) \\
(iii) & f \geq 0 \implies Sf \geq 0
\end{array}
$$

In the discrete setting, property $(i)$ implies that $S$ decomposes as $S = KM$, with $K^\top = K$, while Property $(iii)$ ensures $S_{ij} \geq 0$. Unlike diffusion operators, smoothing operators are not required to preserve constants ($S\mathbf{1} \approx \mathbf{1}$ is possible but not guaranteed) or have eigenvalues bounded by 1, which can lead to numerical vanishing or explosion of signals. Our proposed Sinkhorn normalization allows us to correct such operators into valid diffusion operators.

**Sinkhorn Normalization.** Given a smoothing operator $S$, any positive diagonal scaling $Q = \Lambda S \Lambda$ remains $M$-symmetric and preserves the axioms of Theorem 4.3. Enforcing the property $Q\mathbf{1} = \mathbf{1}$ reduces to finding a single diagonal matrix $\Lambda$ for which the rows of $Q$ sum to one. We show that applying a symmetric variant of the Sinkhorn algorithm recovers both this property and boundedness of the spectrum, thus defining a valid diffusion operator:

**Theorem 4.4** (Symmetric Normalization). *Let* $\mu = \sum_{i=1}^N m_i \delta_{x_i}$ *be a finite discrete measure with positive weights* $m_i > 0$, *and* $S$ *a* smoothing *operator encoded as an N-by-N matrix with positive coefficients* $S_{ij} > 0$. *Then, there exists a unique* diagonal *matrix* $\Lambda$ *with positive*

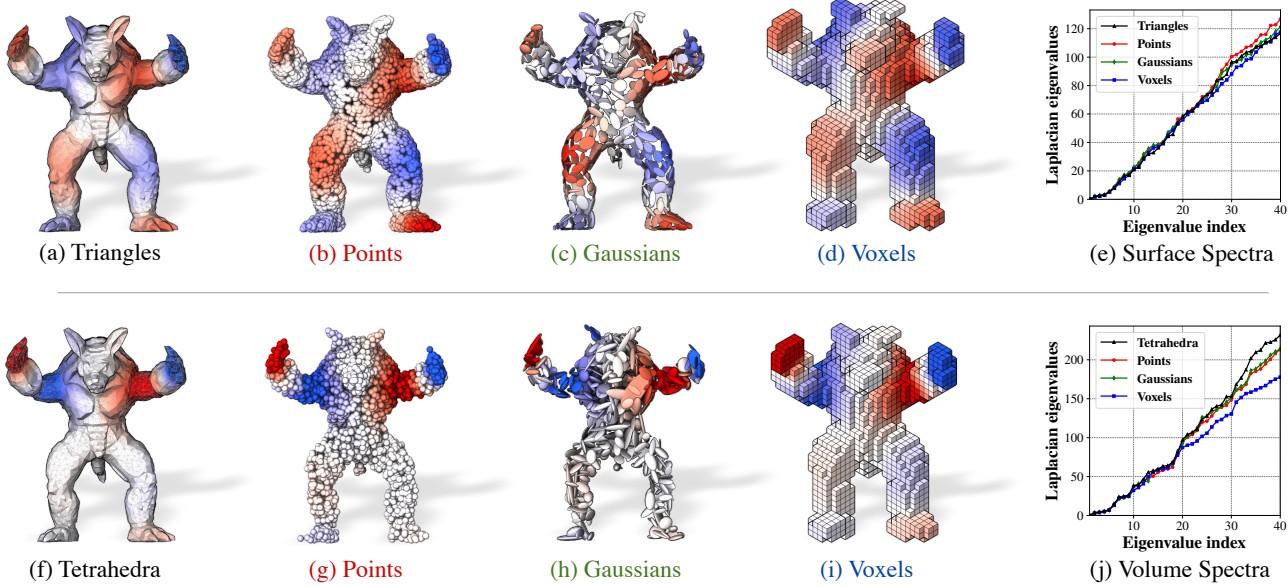

*Figure 3.* Spectral analysis on the Stanford Armadillo (Krishnamurthy & Levoy, 1996) normalized to the unit sphere, treated as a surface (top) and volume (bottom). We compare the reference cotan Laplacian (a,f) to our normalized diffusion operators on clouds of 5 000 points (b,g), mixtures of 500 Gaussians (c,h) and binary voxel masks (d,i), all using a Gaussian kernel of radius $\sigma = 0.05$ (edge length of a voxel). We display the 10th eigenvector (a–d,f–i) and the first 40 eigenvalues (e,j).

coefficients such that $Q = \Lambda S \Lambda$ *is a diffusion operator with respect to $\mu$, which can be obtained using Algorithm 1.*

The proof can be found in Section B. The standard Sinkhorn algorithm (Sinkhorn & Knopp, 1967) alternates row and column normalizations to obtain *distinct* scalings $\Lambda_1, \Lambda_2$ such that $\Lambda_1 S \Lambda_2$ is bi-stochastic, thus breaking the $M$-symmetry. In contrast, the symmetric variant we use (Knight et al., 2014) finds a *single* scaling vector $\lambda$ via $\lambda^{(k+1)} = \sqrt{\lambda^{(k)} \oslash (S\lambda^{(k)})}$, where $\oslash$ denotes element-wise division.

A natural concern is whether this discrete construction remains stable as the sampling density increases. Our second result confirms this for common kernels:

**Theorem 4.5** (Convergence under Refinement). *Let $\mathcal{X} \subset \mathbb{R}^d$ be a bounded domain and $(\mu^s)_{s \in \mathbb{N}}$ be a sequence of finite discrete measures $\mu^s = \sum_i m_i^s \delta_{x_i^s}$ converging weakly to a (potentially continuous) Radon measure $\mu$ with positive, finite total mass. Let $k$ be a Gaussian or exponential kernel with radius $\sigma > 0$, $S^s$ the associated smoothing operators, and $Q^s = \Lambda^s S^s \Lambda^s$ their symmetric normalizations. In particular, $S_{ij}^s = k(x_i^s, x_j^s) m_j^s$.*

*Then, there exist continuous positive functions $\lambda^s, \lambda : \mathcal{X} \to \mathbb{R}_{>0}$ defining the diagonal factors $\Lambda^s$, i.e. $\Lambda_{ii}^s = \lambda^s(x_i^s)$, and such that the integral operator*

$$Qf(x) := \lambda(x) \int_{\mathcal{X}} k(x, y) \lambda(y) f(y) \, \mathrm{d}\mu(y) \quad (2)$$

*is a diffusion operator, and $Q^s$ converges pointwise to $Q$.*

*For all continuous signal $f$ on $\mathcal{X}$,*

$$Q^s f \xrightarrow{s \to \infty} Qf \text{ uniformly on } \mathcal{X}. \quad (3)$$

In other words, our construction is *stable under increasing sampling density*. Unlike un-normalized smoothing operators whose spectral norm can explode with resolution, our normalized operators $Q^s$ converge to a bounded continuous operator $Q$ at a fixed kernel scale $\sigma > 0$ (proof in Section C). The manifold-learning literature usually studies the joint limit $N \to \infty$, $\sigma \to 0$ to recover an underlying Laplace-Beltrami operator (Wormell & Reich, 2021; Cheng & Landa, 2024), and their analysis often passes through a fixed-bandwidth setting as part of a bias-variance decomposition. However, their target is the operator $\frac{1}{\sigma^2}(I - Q)$, which converges to a differential operator. In contrast, we focus on $Q$ itself, as a bounded diffusion operator at the fixed scale used in practice, defined on general discrete domains where no underlying Laplacian is available. Note that we assume finite samples and positive entries in $S$, and refer to Knight et al. (2014, Sec. 3.1) for the case of zero entries.

**Robustness to Mass Perturbation.** A practical concern when using discrete data is the sensitivity of the operator to noise in the estimated mass matrix $M$. The following proposition ensures that symmetric Sinkhorn normalization is *stable*: errors in the mass matrix are damped by a factor of at least 2 in the scaling factors, preventing numerical explosion. We validate this linear relationship experimentally in Section D.

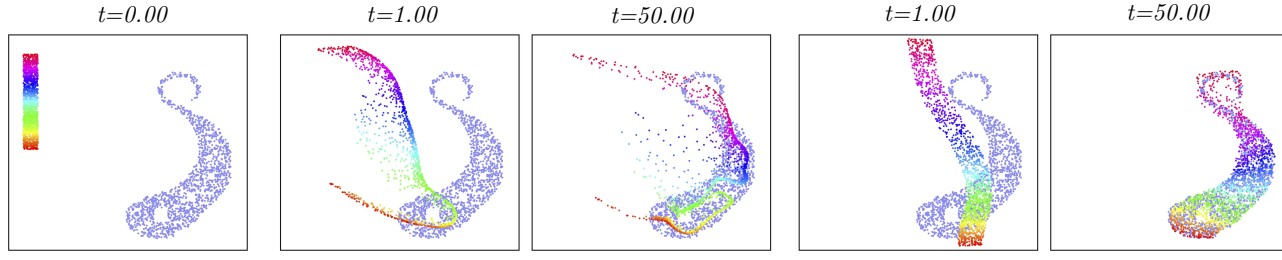

*(a)* Initial State      *(b)* Kernel Smoothing of the Gradient      *(c)* With Normalized Diffusion

*Figure 4.* Flow of a source distribution of points (rainbow) towards a target (blue), following the gradient of the Energy Distance for a Gaussian kernel metric (b) and its normalized counterpart (c).

**Proposition 4.6** (Stability to Mass Perturbation). *Let $Q = \Lambda K M \Lambda$ be a normalized diffusion operator. Consider a first-order relative perturbation of the mass matrix $dm/m$. The induced relative error in the scaling factors $d\lambda/\lambda$ satisfies*

$$\left\| \frac{d\lambda}{\lambda} \right\|_M \leq \frac{1}{2} \left\| \frac{dm}{m} \right\|_M. \tag{4}$$

*Furthermore, the operator variation $dQ$ depends linearly on the mass perturbation $dm/m$.*

Proof can be found in Section D.

## 5. Efficient Implementation

**Sinkhorn Convergence and Versatility.** We use the symmetric Sinkhorn algorithm (Knight et al., 2014; Feydy et al., 2019), which converges in 5-10 iterations in practice (Section E). We show (Section E) this is equivalent to applying standard symmetric scaling to the matrix $MKM$, where convergence is controlled by the geometry of $K$, remaining robust to variations in the mass matrix $M$. While standard Sinkhorn updates can suffer from numerical issues or failure to converge for $\sigma \to 0$, the symmetric variant guarantees stable convergence for any bandwidth (Knight et al., 2014). In the limit $\sigma \to 0$, the normalized matrix simply converges to identity. We treat $S$ as a black-box matrix–vector product: no factorization or complex data structure is required as diffusion behavior is encoded entirely in the diagonal $\Lambda$. Implementation of the operators on CPU and GPU is available at github.com/RobinMagnet/SinkhornKernels.

**Graphs.** Given a symmetric adjacency matrix $A \geq 0$, regularize $A_\varepsilon = A + \varepsilon 11^\top$ ($\varepsilon > 0$). With vertex masses $M = \text{diag}(m_i)$ and degree matrix $D$, set $S = (D + A_\varepsilon)M$, which satisfies Theorem 4.3 and is efficient when $A$ is sparse. Weak diagonal dominance (Horn & Johnson, 1985) ensures a positive spectrum.

**Point Clouds and Gaussian Mixtures.** Given a weighted point cloud $(x_i, m_i)$, we use Gaussian kernels $k$ with $S_{ij} = k(x_i, x_j)m_j$ for smoothing. Matrix-vector products scale to millions of points via the KeOps library (Charlier et al., 2021; Feydy et al., 2020) or optimized attention layers (Lefaudeux et al., 2022; Dao, 2023). We describe how to adapt attention to euclidean distances in Section E

For Gaussian Mixtures, given a pair $m_i \mathcal{N}(x_i, \Sigma_i)$ and $m_j \mathcal{N}(x_j, \Sigma_j)$, we define the pairwise covariance $\mathbf{C}_{ij} := \sigma^2 I + \Sigma_i + \Sigma_j$. We then use use the $L^2$ dot product of densities convolved with an isotropic Gaussian of variance $\sigma^2/2$:

$$S_{ij} = m_j \exp\left[ -\tfrac{1}{2}(x_i - x_j)^\top \mathbf{C}_{ij}^{-1}(x_i - x_j) \right]. \tag{5}$$

Multiplicative constants are normalized out by Algorithm 1.

**Voxel Grids.** On regular grids, Gaussian smoothing is implemented as a *separable* convolution. For sparse volumes, we leverage the efficient data structures of the Taichi library (Hu et al., 2019).

## 6. Results

**Spectral Shape Analysis.** As discussed in Section 4, we expect the leading eigenvectors of a diffusion operator $Q$ to approximate low-frequency Laplacian modes. While spectral convergence typically require vanishing bandwidth (Wormell & Reich, 2021), our construction preserves spectral consistency across diverse representations even at the *fixed, non-zero scales* required for practical geometry processing. Figure 3 compares our operators on different modalities to FEM Laplacian on meshes, and we display additional results in Section G. We estimate Laplacian eigenvalues from our normalized diffusion operators (App. G) and show in Figure 3 (Right) that their distributions remain consistent across modalities, with deviations on volumetric representations emerging near the voxel sampling scale. Additional experiments displayed in Section G highlight stability to change of sampling density and bandwidth parameter $\sigma$.

**Normalized Metrics.** Laplacians naturally induce Sobolev metrics and simple elastic penalties, which are

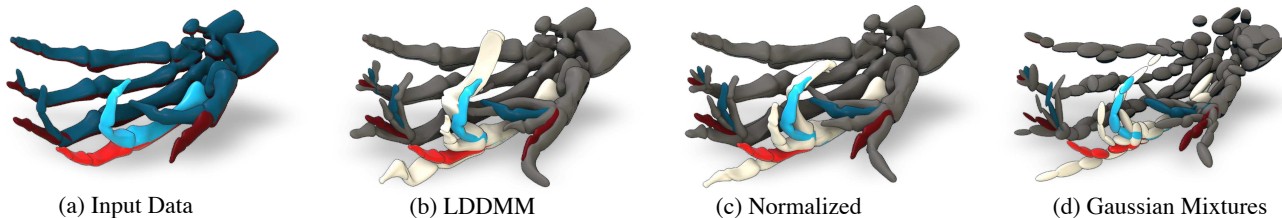

(a) Input Data      (b) LDDMM      (c) Normalized      (d) Gaussian Mixtures

*Figure 5.* **Pose interpolation and extrapolation.** (a) We interpolate between source ($t = 0$, red) and target ($t = 1$, blue) poses, and extrapolate to $t = -0.5$, 0.5, and 1.5. (b) The standard LDDMM geodesic with a Gaussian kernel ($\sigma = 0.1$) produces unrealistic extrapolations. (c) Using our normalized diffusion yields a smoother, more plausible path. (d) The method remains robust on coarse Gaussian-mixture inputs (100 components).

commonly used as regularizers in machine learning and applied mathematics.

Following Feydy et al. (2019), Figure 4 minimizes the Energy Distance (Rizzo & Székely, 2016) between point clouds using gradient smoothing. With standard Gaussian smoothing, each point is simply driven by the raw sum of the neighboring attractions. Points in sparse regions such as boundaries therefore evolve more slowly, and lag behind in the flow. In contrast, our normalized diffusion computes a proper local average. This improves stability near boundaries and significantly accelerates convergence of the Energy–Distance flow, as quantified by the Chamfer distance in Section H. This suggests applications in inverse rendering (Nicolet et al., 2021; Tojo & Umetani, 2025) or generative modelling (Liu & Wang, 2016; Arbel et al., 2019; Korba et al., 2024). Details can be found in Section H.

In Figure 5, following Kilian et al. (2007), we perform geodesic interpolation of anatomical poses. While standard mesh metrics like ARAP (Sorkine & Alexa, 2007) or elastic shell models (Grinspun et al., 2003; Sassen et al., 2024) lack robustness to topological noise, computational anatomy instead use kernel metrics (Bookstein, 1989; Pennec et al., 2019) via LDDMM (Beg et al., 2005; Durrleman et al., 2014). For large deformations, however, kernel metrics favor contraction–expansion dynamics over translations (Micheli et al., 2012) (Figure 5b). Our normalized diffusion $Q$ mitigates this effect, providing more plausible paths across data structures (Figure 5c–d). Intuitively, this occurs because when points cluster, the raw kernel $K$ develops a dominant eigenvalue approaching $n$ that makes the geodesic energy collapse along contraction directions. In contrast, our normalization bounds the spectrum within $[0, 1]$, removing this degeneracy. We verify this effect by plotting the average area distortion across the geodesic path in Section I, along with full equations, implementation details, and a second example on animal meshes.

**Runtimes.** Our symmetric Sinkhorn converges in 5–10 iterations across modalities (see curves in Section E). We report *GPU runtimes* for 5 iterations using a Gaussian kernel for point clouds of increasing size. These are compared to

*Table 1.* Wall-clock runtimes in ms for GPU symmetric Sinkhorn normalization (5 iterations) compared to CPU runtimes for implicit Laplacian diffusion. Dense GPU solvers *exceed memory* at beyond 10k points.

| $N$ | GPU Sinkhorn | CPU LU |
|---|---|---|
| 10,000 | 3 | 65 |
| 50,000 | 21 | 393 |
| 100,000 | 89 | 1,030 |
| 250,000 | 448 | 3,510 |
| 500,000 | 1,817 | 9,100 |
| 1,000,000 | 6,789 | 23,600 |

*CPU runtimes* for implicit Laplacian diffusion using sparse LU factorization, which reflects typical usage *when a Laplacian is available*. Dense solvers on the GPU are significantly slower and run out of memory beyond 10k points. This comparison illustrates practical bottlenecks as sparse direct solvers lack mature GPU implementations. Our approach enables normalizing smoothing operators *at run time* on GPUs for deep learning pipelines. We provide a benchmark of CPU-only implementations in Section F, alongside full details on hardware and baselines.

**Point Feature Learning.** Building on Diffusion-Net (Sharp et al., 2022), we replace its spectral smoothing with our kernel-based operator (*Q-DiffNet*), operating directly on 3D coordinates and learning diffusion scales ($\sigma_i$) instead of diffusion times. We integrate Q-DiffNet into the state of the art ULRSSM pipeline (Cao et al., 2023), train on remeshed FAUST+SCAPE datasets (Bogo et al., 2014; Anguelov et al., 2005; Ren et al., 2019) using point clouds, and also evaluate on SHREC19 (Melzi et al., 2019).

Focusing on *feature extraction*, this experiment isolates the contribution of the diffusion operator. In particular, to ensure fair comparison and prevent variation attributable to input quality, all methods (including point cloud baselines) are provided with the same mesh-derived WKS descriptors. We compare to reference mesh-based methods (Sharp et al., 2022; Cao et al., 2023) as *topology-aware upper bounds*, and their point-cloud retrainings ("PC") as direct competitors.

Note that building a fully Laplacian-free state-of-the-art pipeline would require jointly redesigning inputs, losses, and architecture, and is left to future work.

As shown in Table 2, Q-DiffNet performs on par or better than with point-based baselines on FAUST and SCAPE, and significantly outperforms them on SHREC19, where missing parts can bias spectral diffusion. The method notably outperforms mesh-based upper bounds on SHREC19. As ULRSSM functional-map block still relies on the truncated Laplacian spectrum, we additionally report a *Q-FM* variant that uses the estimated Laplacian spectrum obtained from operator (see Section G). While this removes all mesh-dependent components from the pipeline, performance slightly decreases due to the simple approximation scheme. An ablation using raw XYZ inputs, provided in Section J, confirms that Q-DiffNet remains highly competitive in a strict point-based setting.

Beyond quantitative performance, our framework offers great flexibility. While baselines rely on specific Laplacian designs, our operators apply to arbitrary modalities such as Gaussian splats or voxel grids, and can extend to partial data (Attaiki et al., 2021). This unlocks the application of DiffusionNet to a broader class of geometric data. Implementation details and qualitative results can be found in Section J.

*Table 2.* Mean geodesic error of Q-DiffNet for shape correspondence on FAUST, SCAPE, and SHREC19 (*lower is better*).

|  | Method | FAUST | SCAPE | S19 |
|---|---|---|---|---|
| *Mesh* | DiffNet | **1.6** | 2.2 | 4.5 |
| | ULRSSM | **1.6** | **2.1** | 4.6 |
| *Points* | DiffNet (PC) | 3.0 | 2.5 | 7.5 |
| | ULRSSM (PC) | 2.3 | 2.4 | 5.1 |
| | Q-DiffNet (QFM) | 2.5 | 3.1 | 4.1 |
| | Q-DiffNet | 2.1 | 2.4 | **3.5** |

## 7. Limitations

**Dependence on the Mass Matrix.** Our construction enforces symmetry and mass preservation w.r.t. the inner product defined by $M$. Geometric fidelity thus depends on the quality of $M$. With highly irregular sampling, robust estimation of $M$ can be challenging. While Theorem 4.6 provides stability bounds, large errors in $M$ bias the normalization factors $\Lambda$, distorting the resulting diffusion.

**Strict Mass Preservation.** Preserving mass is not always optimal. In graph processing for instance high-degree nodes might benefit from amplifying features instead of redistributing equally among neighbors. Our normalized operator can limit expressivity in tasks requiring signal re-amplification. However, given recent success of heat-like diffusion in graph

neural networks (Behmanesh et al., 2023; Chamberlain et al., 2021), the choice for information preservation during propagation remains highly application-dependent.

**Theoretical Analysis on Unstructured Data.** While Algorithm 1 provides controlled smoothing across domains, the theoretical link to a continuous Laplace-Beltrami operator is weaker than for mesh or point-cloud setting. Our diffusion operator is a robust drop-in replacement for diffusion, but does not guarantee convergence to an underlying "true" Laplacian.

## 8. Conclusion and Future Works

We introduced a framework for defining heat-like diffusion operators on general geometric data, unifying classical constructions (graph adjacency, similarity matrices) into well-behaved diffusion mechanisms. We demonstrated applications in Laplacian eigenvector approximation, gradient flow stabilization, and integration into neural networks as stable geometry-aware layers.

Future work should explore downstream applications where standard Laplacians are unavailable or unreliable. Scalability can be improved by incorporating sparse or low-rank attention mechanisms for large-scale point clouds and volumetric data.

Additionally, our normalized operator $Q$ naturally fits in the framework of Maas (2011). In this setting, reversible Markov kernels define a Wasserstein-like metric on probability measures, for which heat diffusion coincides with the gradient flow of the entropy. Exploring this connection could enable a more principled usage of $Q$ on general geometric data.

## Acknowledgements

This work has been funded by the RHU ReBone and the France 2030 program managed by the Agence Nationale de la Recherche (ANR-23-IACL-0008, PR[AI]RIE-PSAI). Experiments were performed using HPC resources from GENCI–IDRIS (Grant AD011014760R1).

## Impact Statement

This paper presents work whose goal is to advance the field of Machine Learning. There are many potential societal consequences of our work, none which we feel must be specifically highlighted here.

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

# A. Continuous Formulation of the Metzler Condition

Let $A$ be a real square matrix. We say that $A$ is a *Metzler* matrix if its off-diagonal entries are non-negative. This condition implies that the matrix exponential $e^{tA}$ has non-negative entries for all $t \geq 0$. To see this, remark that for $t$ close enough to 0, we have $e^{tA} = I + tA + o(t)$. This implies that the diagonal coefficients are approximately 1, and the off-diagonal ones are approximately $tA_{ij} \geq 0$. Since for any $t$ we have $e^{tA} = (e^{tA/P})^P$, we can take $P$ large enough such that $e^{tA/P}$ is non-negative (by the small-$t$ argument). Since matrix multiplication preserves non-negativity, $e^{tA}$ is non-negative for all $t \geq 0$.

Reciprocally, if $B$ is a matrix admitting a logarithm $\log(B)$ such that $B^t = \exp(\log(B)t)$ is entrywise positive for all $t \geq 0$, then the identity $B^t = I + t\log(B) + o(t)$ for small $t$ shows that $\log(B)$ is a Metzler matrix.

To extend this reasoning beyond finite-dimensional spaces, we use the formalism of semigroups on Banach spaces (Engel et al., 2000):

**Definition A.1.** Let $T : t \to T(t)$ be a continuous function from $\mathbb{R}$ to the space of bounded linear operators on a Banach space $V$. We say that $T$ is a strongly continuous semigroup if:

    $(i)$    $T(0) = I$
    $(ii)$   $T(t + s) = T(t)T(s)$ for all $t, s \geq 0$
   $(iii)$  $\lim_{t \to 0} T(t)f = f$ for all $f \in V$

Its *generator* $A$ is defined on the set of signals $f \in V$ for which the limit exists as:

$$Af = \lim_{t \to 0} \frac{T(t)f - f}{t} \tag{6}$$

The semigroup is said to be *positive* if $T(t)f \geq 0$ for all $f \geq 0$ and $t \geq 0$.

Under this framework, a Metzler matrix $A$ is the generator of a positive semigroup $t \mapsto e^{tA}$.

In full generality, extending the Metzler condition to infinite-dimensional operators is not straightforward since the notion of "off-diagonal" terms is ill-defined. The correct formalism uses Banach lattices; we refer to (Schaefer, 1974) for proper statements and (Arendt, 1984) for proofs. In our case, we make a simplifying assumption and restrict ourselves to Hilbert spaces of the form $L^2_\mu(\mathcal{X})$, which include both finite-dimensional Euclidean spaces and infinite-dimensional $L^2$ spaces. For any signal $f \in L^2_\mu(\mathcal{X})$, we define $\mathrm{sign}(f)$ pointwise as:

$$\mathrm{sign}(f)(x) = \begin{cases} +1 & \text{if } f(x) > 0 \\ -1 & \text{if } f(x) < 0 \ , \\ 0 & \text{if } f(x) = 0 \end{cases} \quad \text{so that} \quad |f| = \mathrm{sign}(f)f. \tag{7}$$

This leads to the following proposition, which characterizes the Metzler property on general $L^2_\mu(\mathcal{X})$ spaces via a pointwise inequality:

**Proposition A.2.** *Let $A : \mathbb{R}^N \to \mathbb{R}^N$ be a be a linear operator represented by a matrix. Then the following inequalities are equivalent:*

    $(i)$    *(**Metzler condition**) $A_{ij} \geq 0$ whenever $i \neq j$*
    $(ii)$   *(**Kato's inequality**) $A|f| \geq \mathrm{sign}(f)Af$ for all $f \in V$*

*Proof.* Statement $(ii)$ can be rewritten as: for all $i$,

$$\sum_j A_{ij}|f_j| \ \geq \ \mathrm{sign}(f_i)\sum_j A_{ij}f_j \ . \tag{8}$$

If $(i)$ holds, then for $i \neq j$ we have $A_{ij}|f_j| \geq A_{ij}f_j\,\mathrm{sign}(f_i)$, and, by definition, $A_{ii}|f_i| = A_{ii}f_i\,\mathrm{sign}(f_i)$. Therefore we have Equation (8) and $(ii)$.

Conversely, suppose $(ii)$ holds. Consider a pair $i \neq j$ and a signal $f$ such that $f_i = 1$, $f_j = -1$ and $f_k = 0$ for other indices $k$. Equation (8) implies that:

$$A_{ii} + A_{ij} \ \geq \ A_{ii} - A_{ij} \ , \quad \text{i.e.} \quad A_{ij} \ \geq \ 0 \ . \tag{9}$$

This allows us to conclude. □

We would like to extend the implication from the finite-dimensional case: if $A$ satisfies Kato's inequality, then it should generate a semigroup of non-negative operators. In the infinite-dimensional setting, this implication requires additional structure.

**Definition A.3.** A *strictly positive subeigenvector* of an operator $A$ is a function $f \in D(A)$ so that:

$(i)$    $Af \leq \lambda f$ for some $\lambda \in \mathbb{R}$
$(ii)$    $f > 0$ almost everywhere

where $D(A)$ denotes the domain of the (possibly unbounded) operator $A$.

This allows us to state the following result, which is a direct consequence of Theorem 1.7 in Arendt (1984):

**Proposition A.4.** *Let $A$ be a generator of a strongly continuous semigroup on $L_\mu^2(\mathcal{X})$. Assume that there exists a function $g \in D(A)$ such that:*

$(i)$    *$g$ is a strictly positive subeigenvector of $A^{\top_\mu}$.*
$(ii)$    *(weak Kato's inequality) $\langle A^{\top_\mu} g, |f| \rangle_\mu \geq \langle \mathrm{sign}(f)Af, g \rangle_\mu$ for all $f \in D(A)$ .*

*Then the semi-group is positive (see Theorem A.1).*

In our setting, the generator $A$ is equal to the opposite $-\Delta$ of a Laplace-like operator, and $g$ is the constant function $1$. Since our set of axioms implies that $-\Delta^{\top_\mu} 1 = -\Delta 1 = 0$, we always have that $1$ is a strictly positive subeigenvector of $-\Delta^{\top_\mu}$. This allows us to propose the following definition of a Laplace-like operator, which generalizes Theorem 4.1 to discrete measures:

**Definition A.5** (General Laplace-like Operators). Let $\Delta$ be a generator of a strongly continuous semigroup on $L_\mu^2(\mathcal{X})$, where $\mu$ has finite total mass.
We say that $\Delta$ is a *Laplace-like operator* if for all $f \in L_\mu^2(\mathcal{X})$:

$(i)$   **Symmetry:** $\Delta^{\top_\mu} = \Delta$        $(iii)$   **Positivity:** $\langle f, \Delta f \rangle_\mu \geq 0$
$(ii)$   **Constant cancellation:** $\Delta 1 = 0$        $(iv)$   **Kato's inequality:** $\langle \mathrm{sign}(f)\Delta f, 1 \rangle_\mu \geq 0$

The results above show that if $t \mapsto T(t)$ is the strongly continuous semigroup generated by such a Laplace-like operator $\Delta$, then $T(t)$ satisfies the conditions of a diffusion operator (Definition 4.2 in the main manuscript).

We note that the assumption of finite total mass for $\mu$ ensures that the constant function $1$ belongs to $L_\mu^2(\mathcal{X})$, and that our definition includes, as a special case, the classical Laplace–Beltrami operator on compact Riemannian manifolds.

## B. Proof of Theorem 4.4

Our theoretical analysis relies on ideas developed in the context of entropy-regularized optimal transport. We refer to the standard textbook from Peyré & Cuturi (2019) for a general introduction, and to Feydy et al. (2019) for precise statements of important lemmas. Let us now proceed with our proof of Theorem 4.4.

*Proof.* Recall that $\mu = \sum_{i=1}^{N} m_i \delta_{x_i}$ is a finite discrete measure with positive weights $m_i > 0$. The smoothing operator $S$ can be written as the product:

$$S = KM , \tag{10}$$

where $K$ is a $N$-by-$N$ symmetric matrix with positive coefficients $K_{ij} > 0$ and $M = \mathrm{diag}(m_1, \ldots, m_N)$ is a diagonal matrix. Our hypothesis of *operator positivity* on $S$ implies that $K$ is a positive semi-definite matrix. Finally, we can suppose that $\mu$ is a probability measure without loss of generality: going forward, we assume that $m_1 + \cdots + m_N = 1$.

**Optimal Transport Formulation.** We follow Eq. (1) in Feydy et al. (2019) and introduce the symmetric entropy-regularized optimal transport problem:

$$\mathrm{OT}_{\mathrm{reg}}(\mu, \mu) = \min_{\pi \in \mathrm{Plans}(\mu, \mu)} \sum_{i,j=1}^{N} \pi_{ij} C_{ij} + \mathrm{KL}(\pi, mm^\top) \tag{11}$$

where $C_{ij} = -\log K_{ij}$ is the symmetric $N$-by-$N$ *cost* matrix and $\mathrm{Plans}(\mu,\mu)$ is the simplex of $N$-by-$N$ *transport plans*, i.e. non-negative matrices whose rows and columns sum up to $m = (m_1, \ldots, m_N)$. KL denotes the Kullback-Leibler divergence:

$$\mathrm{KL}(\pi, mm^\top) = \sum_{i,j=1}^{N} \pi_{ij} \log \frac{\pi_{ij}}{m_i m_j} \,. \tag{12}$$

Compared with Feydy et al. (2019), we make the simplifying assumption that $\varepsilon = 1$ and do not require that $C_{ii} = 0$ on the diagonal since this hypothesis is not relevant to the lemmas that we use in our paper.

**Sinkhorn Scaling.** The above minimization problem is strictly convex. The fundamental result of entropy-regularized optimal transport, stated e.g. in Feydy et al. (2019, Section 2.1) and derived from the Fenchel-Rockafellar theorem in convex optimization, is that its unique solution can be written as:

$$\pi_{ij} = \exp(f_i + g_j - C_{ij}) \, m_i m_j \,, \tag{13}$$

where $f = (f_1, \ldots, f_N)$ and $g = (g_1, \ldots, g_N)$ are two dual vectors, uniquely defined up to a common additive constant (a pair $(f, g)$ is solution if and only if the pair $(f - c, g + c)$ is also solution) (Feydy et al., 2019, Proposition 11). In our case, by symmetry, there exists a unique constant such that $f = g$ (Feydy et al., 2019, Section B.3). We denote by $\ell = (\ell_1, \ldots, \ell_N)$ this unique "symmetric" solution. It is the unique vector such that:

$$\pi_{ij} = \exp(\ell_i + \ell_j - C_{ij}) \, m_i m_j = m_i e^{\ell_i} \, K_{ij} \, e^{\ell_j} m_j \tag{14}$$

is a valid transport plan in $\mathrm{Plans}(\mu,\mu)$. This matrix is symmetric and such that for all $i$:

$$\sum_{j=1}^{N} \pi_{ij} = m_i \qquad \text{i.e.} \qquad e^{\ell_i} \sum_{j=1}^{N} K_{ij} \, e^{\ell_j} m_j = 1 \,. \tag{15}$$

We introduce the positive scaling coefficients $\lambda_i = e^{\ell_i}$, the diagonal scaling matrix $\Lambda = \mathrm{diag}(\lambda_1, \ldots, \lambda_N)$, and rewrite this equation as:

$$\Lambda K M \Lambda 1 = 1 \qquad \text{i.e.} \qquad Q1 = 1 \text{ where } Q = \Lambda K M \Lambda \,. \tag{16}$$

This shows that scaling $S = KM$ with $\Lambda$ enforces our **constant preservation** axiom for diffusion operators – property $(ii)$ in Theorem 4.2. Likewise, since $\Lambda$ is a diagonal matrix with positive coefficients, $Q$ satisfies axioms $(i)$ – **symmetry** with respect to $M$ – and $(iv)$ – **entrywise positivity**.

Crucially, $\Lambda$ can be computed efficiently using a symmetrized Sinkhorn algorithm: our Algorithm 1 is directly equivalent to in Feydy et al. (2019, Eq. (25)).

**Spectral Normalization.** To conclude our proof, we now have to show that the normalized operator $Q$ also satisfies axiom $(iii)$ – **damping** – in our definition of diffusion operators, i.e. show that its eigenvalues all belong to the interval $[0, 1]$.

To this end, we first remark that $Q = \Lambda K M \Lambda = \Lambda K \Lambda M$ has the same eigenvalues as $Q' = \sqrt{M} \Lambda K \Lambda \sqrt{M}$, where $\sqrt{M} = \mathrm{diag}(\sqrt{m_1}, \ldots, \sqrt{m_N})$. If $\alpha$ is a scalar and $x$ is a vector, the eigenvalue equation:

$$Qx = \Lambda K \Lambda \sqrt{M} \underbrace{\sqrt{M} x}_{y} = \alpha x \qquad \text{is equivalent to} \qquad Q'y = \sqrt{M} \Lambda K \Lambda \sqrt{M} y = \alpha y \tag{17}$$

with the change of variables $y = \sqrt{M} x$.

Then, we remark that for any vector $x$ in $\mathbb{R}^N$,

$$\sum_{i,j=1}^{N} K_{ij} \lambda_i \lambda_j (\sqrt{m_j} x_i - \sqrt{m_i} x_j)^2 \tag{18}$$

$$= \sum_{i,j=1}^{N} K_{ij} \lambda_i \lambda_j \left( m_j x_i^2 + m_i x_j^2 - 2\sqrt{m_i}\sqrt{m_j} x_i x_j \right) \tag{19}$$

$$= \sum_{i=1}^{N} \underbrace{(\Lambda K M \Lambda 1)_i}_{Q1=1} x_i^2 \; + \; \sum_{j=1}^{N} \underbrace{(\Lambda K M \Lambda 1)_j}_{Q1=1} x_j^2 \; - \; 2\, x^\top Q' x \tag{20}$$

$$= 2\, x^\top (I - Q') x \,. \tag{21}$$

Since the upper term is non-negative as a sum of squares, we get that the eigenvalues of the symmetric matrix $I - Q'$ are all non-negative. This implies that the eigenvalues of $Q'$, and therefore the eigenvalues of $Q$, are bounded from above by 1.

In the other direction, recall that our hypothesis of operator positivity on $S$ implies that $K$ is a positive semi-definite matrix. This ensures that Q', and therefore Q, also have non-negative eigenvalues. Combining the two bounds, we show that the spectrum of the normalized operator $Q$ is, indeed, included in the unit interval $[0, 1]$. $\qquad \square$

## C. Proof of Theorem 4.5

*Proof.* The hypotheses of Theorem 4.5 fit perfectly with those of Feydy et al. (2019, Theorem 1). Notably, we make the assumption that $\mathcal{X}$ is a bounded region of $\mathbb{R}^d$: we can replace it with a closed ball of finite radius, which is a compact metric space. Just as in Section B, we can assume without loss of generality that the finite measures $\mu^s$ and the limit measure $\mu$ are probability distributions, that sum up to 1: positive multiplicative constants are absorbed by the scaling coefficients $\Lambda^s$ and $\Lambda$.

If $k(x, y)$ is a Gaussian kernel of deviation $\sigma > 0$, we use the cost function $C(x, y) = \frac{1}{2}\|x - y\|^2$ and an entropic regularization parameter $\varepsilon = \sigma^2$. If $k(x, y)$ is an exponential kernel at scale $\sigma > 0$, the cost function is simply the Euclidean norm $\|x - y\|$ and the entropic regularization parameter $\varepsilon$ is equal to $\sigma$.

**Continuous Scaling Functions.** The theory of entropy-regularized optimal transport allows us to interpret the dual variables $f$, $g$ and $\ell$ of Eqs. (13-14) as continuous functions defined on the domain $\mathcal{X}$. Notably, for any probability distribution $\mu$, the continuous function $\ell : \mathcal{X} \to \mathbb{R}$ is uniquely defined by the "Sinkhorn equation" – see Sections B.1 and B.3 in Feydy et al. (2019):

$$\forall x \in \mathcal{X}, \;\; \ell(x) \; = \; -\varepsilon \log \int_{\mathcal{X}} \exp \tfrac{1}{\varepsilon} \big( \ell(y) - C(x, y) \big) \, \mathrm{d}\mu(y) \,. \tag{22}$$

The first part of Theorem 4.2 is a reformulation of this standard result. We introduce the continuous, positive function:

$$\lambda(x) \; = \; \exp(\ell(x)/\varepsilon) \; > \; 0 \tag{23}$$

which is bounded on the compact domain $\mathcal{X}$. We remark that Eq. (22) now reads:

$$\forall x \in \mathcal{X}, \;\; \lambda(x) \; = \; \frac{1}{\int_{\mathcal{X}} k(x, y) \lambda(y) \, \mathrm{d}\mu(y)} \tag{24}$$

$$\text{i.e. } 1 \; = \; \lambda(x) \int_{\mathcal{X}} k(x, y) \lambda(y) \, \mathrm{d}\mu(y) \,. \tag{25}$$

This implies that the operator $Q$ defined in Equation (2) satisfies our **constant preservation** axiom for diffusion operators. By construction, it also satisfies the **symmetry** and **entrywise positivity** axioms. The **damping** property derives from the fact that we can write $Q$ as the limit of the sequence of discrete diffusion operators $Q^s$ with eigenvalues in $[0, 1]$.

**Convergence.**    To prove it, note that the above discussion also applies to the discrete measures $\mu^s = \sum_{i=1}^{N_s} m_i^s \delta_{x_i^s}$. We can uniquely define a continuous function $\ell^s : \mathcal{X} \to \mathbb{R}$ such that:

$$\forall x \in \mathcal{X}, \ \ell^s(s) \ = \ -\varepsilon \log \sum_{j=1}^{N_s} m_j^s \exp \tfrac{1}{\varepsilon}\big(\ell^s(x_j^s) - C(x, x_j^s)\big) , \tag{26}$$

and interpret the diagonal coefficients of the scaling matrix $\Lambda^s$ as the values of the positive scaling function:

$$\lambda^s(x) \ = \ \exp(\ell^s(x)/\varepsilon) \ > \ 0 \tag{27}$$

sampled at locations $(x_1^s, \ldots, x_{N_t}^s)$.

Recall that the sequence of discrete measures $\mu^s$ converges weakly to $\mu$ as $s$ tends to infinity. Crucially, Proposition 13 in Feydy et al. (2019) implies that the dual potentials $\ell^s$ converge *uniformly on* $\mathcal{X}$ towards $\ell$. Since $\ell$ is continuous and therefore bounded on the compact domain $\mathcal{X}$, this uniform convergence also holds for the (exponentiated) scaling functions: $\lambda^s$ converges uniformly on $\mathcal{X}$ towards $\lambda$.

For any continuous signal $f : \mathcal{X} \to \mathbb{R}$, we can write down the computation of $Q^s f$ as the composition of a pointwise multiplication with a scaled positive measure $\mu^s \lambda^s$, a convolution with the (fixed, continuous, bounded) kernel $k$, and a pointwise multiplication with the positive scaling function $\lambda^s$. In other words:

$$Q^s f \ = \ \lambda^s \cdot \big(k \star (\mu^s \lambda^s f)\big) . \tag{28}$$

Likewise, we have that:

$$Qf \ = \ \lambda \cdot \big(k \star (\mu \lambda f)\big) . \tag{29}$$

Since $\lambda^s$ converges uniformly towards $\lambda$ and $f$ is continuous, the signed measure $\mu^s \lambda^s f$ converges weakly towards $\mu \lambda f$. This implies that the convolution with the (bounded) Gaussian or exponential kernel $k \star (\mu^s \lambda^s f)$ converges uniformly on $\mathcal{X}$ towards $k \star (\mu \lambda f)$, which allows us to conclude. $\qquad\square$

# D. Stability to Noise in Mass

**Infinitesimal Variation.**    Let $KM$ be a smoothing operator as in Section 4, with $M = \mathrm{diag}(m)$. Theorem 4.4 guarantees the existence of a diagonal matrix $\Lambda = \mathrm{diag}(\lambda)$ such that $Q = \Lambda K M \Lambda$ is a diffusion operator.

Consider an infinitesimal variation $m \to m + dm$ of $m$. This induces a variation of the scaling factors $\lambda \to \lambda + d\lambda$. To find the relationship between them, we differentiate the row-stochasticity condition $\sum_j \lambda_i \lambda_j k_{ij} m_j = 1$. We obtain

$$\frac{d\lambda_i}{\lambda_i} + \sum_j \lambda_i k_{ij} m_j \lambda_j \frac{d\lambda_j}{\lambda_j} + \sum_j \lambda_i k_{ij} m_j \lambda_j \frac{dm_j}{m_j} = 0. \tag{30}$$

Since $Q_{ij} = \lambda_i k_{ij} m_j \lambda_j$, this simplifies to

$$\frac{d\lambda}{\lambda} + Q\frac{d\lambda}{\lambda} + Q\frac{dm}{m} = 0 \quad \Longrightarrow \quad \frac{d\lambda}{\lambda} = -(\mathrm{Id} + Q)^{-1} Q \frac{dm}{m}. \tag{31}$$

This shows that the relative error in mass propagates to the Sinkhorn scaling factors through the operator $-(\mathrm{Id} + Q)^{-1} Q$.

**Stability Bound.**    We can bound the amplification of this error using the $M$-spectral norm $\|\cdot\|_M$. Since $Q$ is $M$-symmetric with eigenvalues in $(0, 1]$, $A = (\mathrm{Id} + Q)^{-1} Q$ is also $M$-symmetric, and its eigenvalues are in the form $\frac{\mu}{1+\mu}$ where $\mu \in (0, 1]$. Since the function $x \mapsto \frac{x}{1+x}$ is strictly increasing and bounded by $1/2$ on $[0, 1]$, the operator norm is also bounded:

$$\left\|\frac{d\lambda}{\lambda}\right\|_M \leq \frac{1}{2}\left\|\frac{dm}{m}\right\|_M . \tag{32}$$

This proves that symmetric Sinkhorn normalization is *stable*: relative errors in the mass matrix are damped by a factor of at least 2 in the scaling factors, preventing numerical explosion.

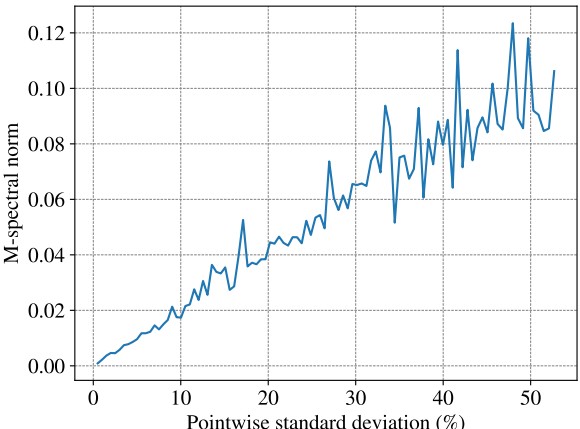

*Figure 6.* Sensitivity of the diffusion operator to mass perturbations. We evaluate the stability of our operator on the Armadillo mesh by applying multiplicative log-normal noise to the mass matrix $M$. The x-axis shows pointwise standard deviation of the noise (in %), and the y-axis the relative error, measured as the $M$-spectral norm of the difference between the unperturbed and perturbed operators ($\|Q_m - Q_{m_\sigma}\|_M$). The error scales linearly with the noise magnitude, empirically confirming that the symmetric Sinkhorn normalization prevents the amplification of mass estimation errors.

**Operator Variation.**  Finally, the first-order variation $dQ$ of $Q$ is given by

$$dQ = \operatorname{diag}\left(\frac{d\lambda}{\lambda}\right) Q + Q \operatorname{diag}\left(\frac{dm}{m}\right) + Q \operatorname{diag}\left(\frac{d\lambda}{\lambda}\right). \tag{33}$$

Using Eq. (31) in this expression shows that variations in $Q$ depend linearly on the relative log-variations of the mass $m$, with coefficients bounded by the spectral properties of $Q$.

**Experiment.**  We test this theoretical result experimentally. We introduce multiplicative log-normal noise to the (lumped) masses of the Armadillo mesh: $(m_\sigma)_i = m_i e^{\epsilon_i} e^{-\sigma^2/2}$, where $\epsilon_i \sim \mathcal{N}(0, \sigma^2)$. The term $e^{-\sigma^2/2}$ ensures the expectation remains unbiased, *i.e.* $\mathbb{E}_{\epsilon_i}\left[(m_\sigma)_i\right] = m_i$.

In Figure 6, we plot the relative error $\|Q_m - Q_{m_\sigma}\|_M$ against the noise level $\sigma$, where $Q_m$ and $Q_{m_\sigma}$ are the diffusion operators associated to $KM$ and $KM_\sigma$ respectively. We use a Gaussian kernel K of bandwidth 0.05. As predicted by our derivation, the error scales linearly with the input noise magnitude and remains well-controlled even for significant perturbations, confirming the robustness of the method to mass estimation errors.

## E. Q-Diffusion in Practice

**Equivalence to Symmetric Sinkhorn.**  Algorithm 1 simply solves the fixed point equation $\Lambda KM\Lambda 1 = 1$. Since $M$ and $\Lambda$ are diagonal, they commute and therefore the problem is equivalent to $\Lambda MKM\Lambda 1 = M1$. This boils down to the standard symmetric Sinkhorn algorithm from (Knight et al., 2014) applied to matrix $MKM$, with marginals $M1 = \operatorname{diag}(M)$. This equivalence explains the fast convergence in 5 to 10 iterations we observe in practice, which was studied in (Knight et al., 2014).

**Sinkhorn Convergence.**  We evaluate the convergence behavior of the symmetrized Sinkhorn algorithm across various settings. Specifically, we monitor the quantity:

$$\frac{\int \left|\Lambda^{(i)} S \Lambda^{(i)} 1 - 1\right| d\mu}{\int d\mu} \tag{34}$$

where $\Lambda^{(i)}$ denotes the diagonal scaling matrix after $i$ Sinkhorn iterations. This corresponds to the average deviation between the constant signal 1 and its smoothed counterpart $Q^{(i)} 1 = \Lambda^{(i)} S \Lambda^{(i)} 1$ on the domain that is defined by the positive measure $\mu$. According to our definition, both signals coincide when $Q^{(i)}$ is a smoothing operator. Figure 7 presents these results,

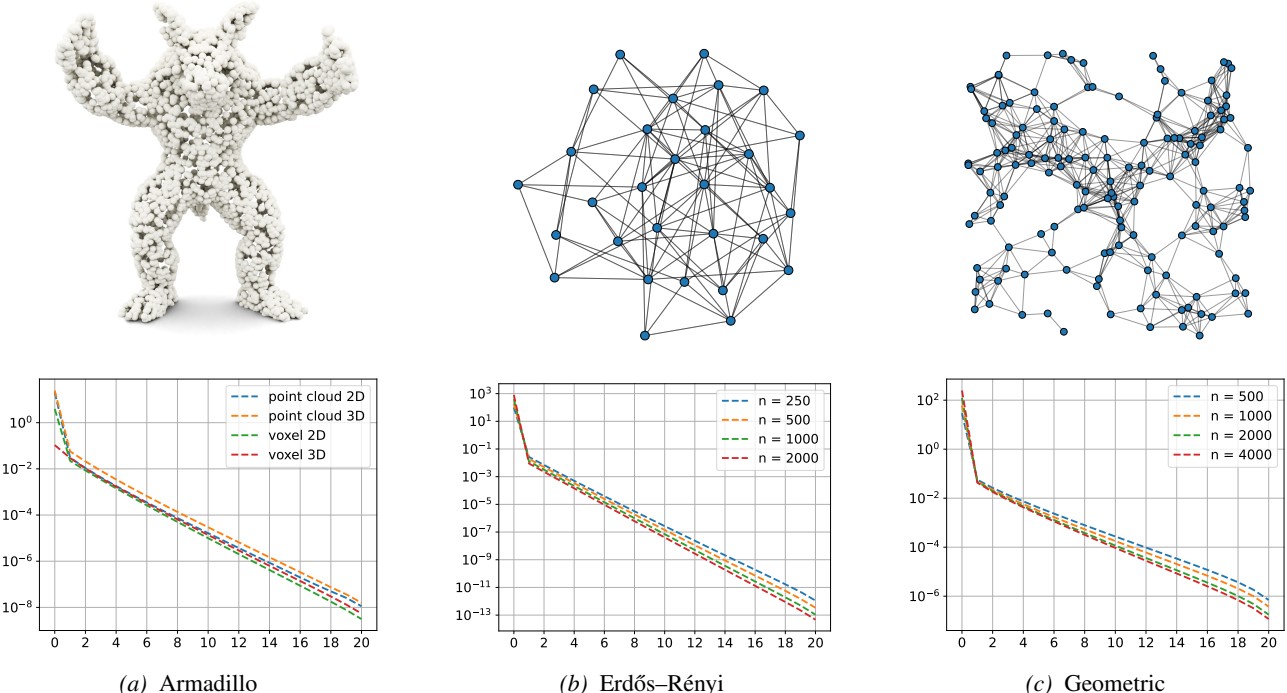

*(a)* Armadillo          *(b)* Erdős–Rényi          *(c)* Geometric

*Figure 7.* Convergence of the symmetric Sinkhorn algorithm on (a) the Armadillo shape, (b) a graph with $N$ nodes and random edges, (c) a random geometric graph with $N$ nodes

with visualizations of the input modalities along the top row and corresponding convergence curves below. In Figure 7a, we report results for different representations of the Armadillo shape used in the main paper: uniform point cloud samples on the surface and volume, as well as voxel-based representations of the boundary and interior. Figure 7b illustrates the behavior on Erdős–Rényi random graphs with edge probability $p = 0.2$, and Figure 7c shows results on geometric graphs, where points are uniformly sampled in the unit square and edges are drawn between points within radius $r = 0.15$.

Across all experiments, we observe rapid convergence: typically, 5 to 10 iterations are sufficient to reach error levels below $10^{-3} = 0.1\%$. We note that one iteration of our algorithm corresponds to the classical symmetric normalization of graph Laplacians, which satisfies our constant preservation property up to a precision of $1\%$ to $5\%$. From this perspective, we understand our work as a clarification of the literature on graph Laplacians. While most practitioners are used to working with *approximate* normalization, we provide a clear and affordable method to satisfy this natural axiom up to an *arbitrary tolerance* parameter. We argue that this is preferable to choosing between row-wise normalization (which guarantees the preservation of constant signals, but discards symmetry) and standard symmetric normalization (which makes a small but noticeable error on the preservation of constant signals).

**Implementing Q-Diffusion on Discrete Samples.** Let $K \in \mathbb{R}^{N \times N}$ be a symmetric kernel matrix, and $M = \text{diag}(m_1, \ldots, m_N)$ be a diagonal mass matrix. As described in Algorithm 1, we compute a diagonal scaling matrix $\Lambda = \text{diag}(e^{\ell_1}, \ldots, e^{\ell_N})$, such that the normalized diffusion operator becomes $Q = \Lambda K M \Lambda$. This operator can be implemented efficiently using the KeOps library (Charlier et al., 2021; Feydy et al., 2020), which avoids instancing the dense kernel matrix.

In the case where $K$ is a Gaussian kernel between points $x_i$ in $\mathbb{R}^d$, with standard deviation $\sigma > 0$, applying $Q$ to a signal $f \in \mathbb{R}^N$ gives:

$$(Qf)_i = \sum_{j=1}^{N} \exp\left(-\tfrac{1}{2\sigma^2} \|x_i - x_j\|^2 + \ell_i + \ell_j\right) m_j f_j \tag{35}$$

$$= \sum_j \exp\left(q_{ij}\right) f_j \quad \text{where } q_{ij} := -\tfrac{1}{2\sigma^2}\|x_i - x_j\|^2 + \ell_i + \ell_j + \log m_j. \tag{36}$$

*Table 3.* CPU runtime benchmark comparing the Implicit Euler solver and KNN-accelerated Sinkhorn. All times are reported in seconds, with KNN precomputation times indicated in parentheses.

| $N$ | **Implicit Euler** | **Sinkhorn ($k = 25$)** | **Sinkhorn ($k = 100$)** |
|---|---|---|---|
| 10,000 | 0.428 | 0.010 (+0.117) | 0.022 (+0.274) |
| 100,000 | 2.921 | 0.064 (+0.612) | 0.338 (+1.819) |
| 1,000,000 | 71.776 | 0.971 (+6.004) | 5.199 (+18.351) |

Since $Q$ is row-normalized by construction, (i.e., $Q1 = 1$), this operation can be written as a softmax-weighted sum:

$$(Qf)_i = \sum_{j=1}^{N} \text{SoftMax}_j \left( q_{ij} \right) f_j \,. \tag{37}$$

We note that the scores $q_{ij}$ can be expressed as inner products between extended embeddings $\tilde{x}_i, \tilde{y}_i \in \mathbb{R}^{d+2}$, enabling fast attention implementations:

$$q_{ij} = \tilde{x}_i^\top \tilde{y}_j \,, \text{ where } \tilde{x}_i = \begin{pmatrix} \frac{1}{\sigma} x_i \\ \ell_i - \frac{1}{2\sigma^2} \|x_i\|^2 \\ 1 \end{pmatrix} \text{ and } \tilde{y}_j = \begin{pmatrix} \frac{1}{\sigma} x_j \\ 1 \\ \ell_j - \frac{1}{2\sigma^2} \|x_j\|^2 + \log(m_j) \end{pmatrix} \,. \tag{38}$$

This leads to an attention-style formulation of the operator:

$$Qf = \text{Attention} \left( \tilde{X}, \tilde{Y}, f \right) \tag{39}$$

where $\tilde{X}, \tilde{Y} \in \mathbb{R}^{N \times (d+2)}$ are the stacked embeddings of all points. This makes $Qf$ compatible with fast attention layers such as FlashAttention (Dao, 2023) or xFormers (Lefaudeux et al., 2022). Note that the softmax normalization in the Attention layer is invariant to additive constants in $\tilde{x}_i$, allowing the implementation to be further simplified using only a $(d + 1)$-dimensional embeddings for $\tilde{X}$ and $\tilde{Y}$

**Spectral Decomposition.** The largest eigenvectors of a symmetric matrix can be efficiently computed using the power method or related iterative solvers (Saad, 2011). However, standard routines typically assume symmetry with respect to the standard inner product. Since our diffusion operator $Q = \Lambda K M \Lambda$ is symmetric with respect to the $M$-weighted inner product, we need to reformulate the problem. Noting that $M$ and $\Lambda$ are diagonal and therefore commute, we can write

$$Q = M^{-1} \left( \Lambda M K M \Lambda \right) \,. \tag{40}$$

This allows us to compute the eigenvectors and eigenvalues of $Q$ by solving the following generalized eigenproblem for symmetric matrices:

$$(\Lambda M K M \Lambda)\Phi = \lambda M \Phi \,. \tag{41}$$

This is supported by standard linear algebra routines, such as `scipy.sparse.linalg.eigsh` (Virtanen et al., 2020), and ensures that the resulting eigenvectors $\Phi$ are orthogonal with respect to the $M$ inner product.

# F. Runtimes

**Setup.** We time **5 iterations** of the symmetric Sinkhorn normalization on point-cloud kernels using PyTorch + PyKeOps (symbolic lazy tensors) on an NVIDIA V100 (CUDA 12.1). For reference, we also time implicit Laplacian diffusion via a sparse LU solve of $(M+t\Delta)$ on an Intel Xeon Gold 6248 CPU. This reflects typical usage: kernel mat–vecs map well to GPUs, whereas sparse direct solvers are mature and memory-efficient on CPUs when a Laplacian is available.

**What is timed.** Sinkhorn: each iteration = one mat–vec with $S$ + a diagonal rescaling; we report wall-clock for 5 iterations. Laplacian: factorization + one solve of $(M+t\Delta)^{-1}b$ on CPU (best-case when a sparse $\Delta$ exists). Our GPU timings use a brute-force kernel on an unstructured 3D point cloud, and could be further improved.

**Dense GPU baselines.** On the GPU, we also measured a Cholesky factorization (210 ms) and a matrix exponential (770 ms) of a Laplacian in PyTorch and 10000 vertices. These approaches exceed GPU memory limits beyond ~10k points.

**CPU Runtimes.** When running on CPU, standard dense Sinkhorn can become prohibitively slow with large number of samples. A standard strategy, in particular in our setting, is to use a precomputed K-nearest neighbor graph and run Sinkhorn on a sparse matrix with only $NK$ elements. On Table 3, we display CPU runtimes for Implicit Euler and sparse Sinkhorn using $k = 25$ and $k = 100$ neighbors. Evaluations were performed on a dual-socket Intel Xeon E5-2695 v4, and show that sparse sinkhorn remains significantly faster than Implicit Euler.

**Sinkhorn Complexity.** Per Sinkhorn iteration for different methods:

 (i) **Dense matrices** $S$: $O(N^2)$ time/memory
 (ii) **Symbolic kernel** $S$ (e.g., Gaussian with PyKeOps): $O(N^2)$ time, $O(N)$ memory sparse $S$ ($k$-NN): $O(kN)$ time (generally $O(\text{nnz})$)
 (iii) **Low-rank** (rank $R$): $O(RN^2)$
 (iv) **Grid convolution**: $O(N)$ for small filters, $O(N \log N)$ for large filters using FFTs

**Baseline Complexity.** Per diffusion step via Laplacian-based methods:

 (i) **Matrix exponential (dense)**: $O(N^3)$ time; rarely used at scale.
 (ii) **Implicit Euler** $(I+t\Delta)^{-1}$ with sparse LU/Cholesky: worst case $O(N^3)$; for mesh-like sparsity typically $O(N^{1.5})$ for factorization and $O(N^2)$ per solve (amortizable across right-hand sides).
 (iii) **Spectral truncation** (rank $R$): $O(RN^2)$ in the dense setting; truncation may introduce ringing artifacts.

These baselines assume access to a well-defined sparse Laplacian and specialized linear algebra routines.

## G. Details on Eigenvectors Computation

**FEM Laplacian on Tetrahedral Meshes.** In Figure 3 of the main paper, we implement our method on different representations of the Armadillo, treated as a surface and as a volume with uniform density. For the surface mesh in Figure 3a, we use the standard cotangent Laplacian as a reference. For the tetrahedral mesh shown in Figure 3f, we use a finite element Laplacian, generalizing the cotangent Laplacian in 2D (Crane, 2019). Let $\{e_i\}$ be a basis of piecewise linear basis and $\{de_i\}$ their gradients. The discrete Laplacian $\Delta$ takes the form:

$$\Delta = M^{-1}L \, , \tag{42}$$

where $L$ is the stiffness matrix and $M$ is the mass matrix defined by:

$$L_{ij} = \langle de_i, de_j \rangle \, , \quad M_{ij} = \langle e_i, e_j \rangle \, . \tag{43}$$

Following Crane (2019), we compute the off-diagonal entries of $L$ with:

$$L_{ij} = \frac{1}{6} \sum_{ijkl \in \mathcal{T}} l_{kl} \cot(\theta_{kl}^{ij}) \, , \tag{44}$$

where $\mathcal{T}$ is the set of tetrahedra in the mesh, $l_{kl}$ is the length of edge $kl$, and $\theta_{kl}^{ij}$ is the dihedral angle between triangles $ikl$ and $jkl$. The diagonal entries are defined to make sure that the rows sum to zero:

$$L_{ii} = -\sum_{j \neq i} L_{ij} \, . \tag{45}$$

The entries of the mass matrix $M$ are given by:

$$M_{ij} = \frac{1}{20} \sum_{ijkl \in \mathcal{T}} \text{vol}(ijkl) \text{ for } i \neq j \, , \quad M_{ii} = \frac{1}{10} \sum_{ijkl \in \mathcal{T}} \text{vol}(ijkl) \, . \tag{46}$$

**Point Clouds.** As discussed in the main paper, we compare the eigendecompositions of these cotan Laplacians to that of our normalized Gaussian smoothings on discrete representations of the Armadillo. For the sake of simplicity, Figure 3b and 3g correspond to uniform discrete samples, i.e. weighted sums of Dirac masses:

$$\mu \;=\; \sum_{i=1}^{N} \tfrac{1}{N}\delta_{x_i}\,, \tag{47}$$

where $x_1, \ldots, x_N$ correspond to $N = 5\,000$ three-dimensional points drawn at random on the triangle mesh (for Figure 3b) and in the tetrahedral volume (for Figure 3g).

**Gaussian Mixtures.** To compute the Gaussian mixture representations of Figures 3c and 3h, we simply rely on the Scikit-Learn implementation of the EM algorithm with K-Means++ initialization (Pedregosa et al., 2011) and 500 components. This allows us to write:

$$\mu \;=\; \sum_{i=1}^{500} m_i\,\mathcal{N}(x_i, \Sigma_i)\,, \tag{48}$$

where the scalars $m_i$ are the non-negative mixture weights, the points $x_i$ are the Gaussian centroids and the 3-by-3 symmetric matrices $\Sigma_i$ are their covariances.

**Mass Estimation on Voxel Grids.** To encode the Armadillo's *volume* as a binary mask in Figure 3i, we simply assign a mass of 1 to voxels that contain points inside of the watertight Armadillo surface. This allows us to demonstrate the robustness of our implementation, even when voxel values do not correspond to the exact volume of the intersection between the tetrahedral mesh and the cubic voxel.

However, this approach is too simplistic when representing the *surface* of the Armadillo with voxels. Since the grid is more densely sampled along the $xyz$ axes than in other directions, assigning a uniform mass of 1 to every voxel that intersects the triangle mesh would lead to biased estimates of the mass distribution. To address this quantization issue, we use kernel density estimation to assign a mass $m_i$ to each voxel.

As described above, we first turn the triangle mesh into a binary mask. Then, for every non-empty voxel $x$, we use an isotropic Gaussian kernel $k$ with standard deviation $\sigma$ equal to 3 voxels to estimate a voxel mass $m(x)$ with:

$$m(x) = \frac{1}{\sum_y k(x,y)} \tag{49}$$

where the sum is taken over neighboring, non-empty voxels.

**Estimation of the Laplacian Eigenvalues.** Recall that with our conventions, the Laplace operator is non-negative. In both of our settings (surface and volume), performing an eigendecomposition of the reference cotan Laplacian yields an *increasing* sequence of eigenvalues starting at $\lambda_1^\Delta = 0$. On the other hand, computing the largest eigenvalues of a normalized diffusion operator yields a *decreasing* sequence of eigenvalues starting at $\lambda_1^Q = 1$.

To compare both sequences with each other and produce the curves of Figure 3e and 3j, we propose the following simple heuristics for diffusions $Q$ derived from a Gaussian kernel of deviation $\sigma > 0$:

- For point clouds and voxels, we use:

$$\lambda_i \;=\; -\frac{2}{\sigma^2}\log(\lambda_i^Q)\,. \tag{50}$$

  Indeed, when the underlying measure $\mu$ corresponds to a regular grid with uniform weights, we can interpret the Gaussian kernel matrix as a convolution operator with a Gaussian kernel $\exp(-\|x\|^2/2\sigma^2)$. Its eigenvalues can be computed in the Fourier domain as $\exp(-\sigma^2\|\omega\|^2/2)$. To recover the eigenvalues $\|\omega\|^2$ of the Laplace operator, we simply have to apply a logarithm and multiply by $-2/\sigma^2$.

- For Gaussian mixtures with component weights $m_i$, centroids $x_i$ and covariance matrices $\Sigma_i$, we use:

$$\lambda_i \;=\; -\frac{2}{\sigma^2 + \frac{2}{d}(\sum_i m_i\mathrm{trace}(\Sigma_i))/(\sum_i m_i)}\log(\lambda_i^Q)\,, \tag{51}$$

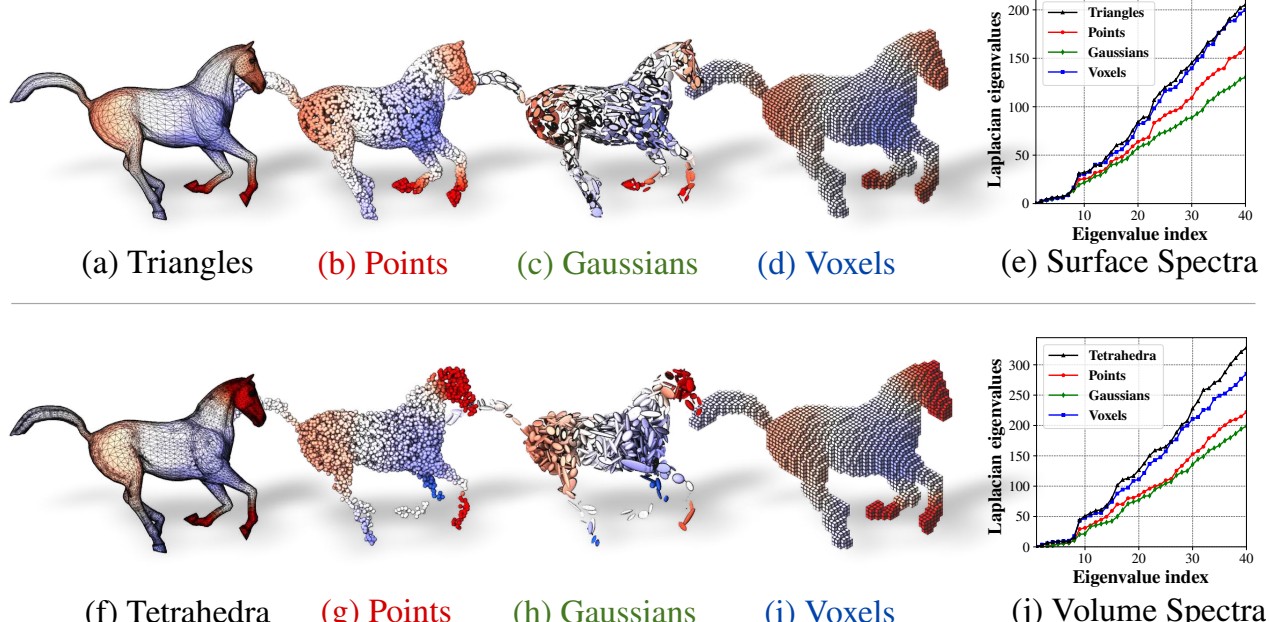

(a) Triangles  (b) Points  (c) Gaussians  (d) Voxels  (e) Surface Spectra

(f) Tetrahedra  (g) Points  (h) Gaussians  (i) Voxels  (j) Volume Spectra

*Figure 8.* Spectral analysis on the galloping horse (Sumner & Popović, 2004) normalized to the unit sphere, treated as a surface (top) and volume (bottom). We compare the reference cotan Laplacian (a,f) to our normalized diffusion operators on clouds of 5 000 points (b,g), mixtures of 500 Gaussians (c,h) and binary voxel masks (d,i), all using a Gaussian kernel of radius $\sigma = 0.05$ (edge length of a voxel). We display the 8 th eigenvector (a–d,f–i) and the first 40 eigenvalues (e,j).

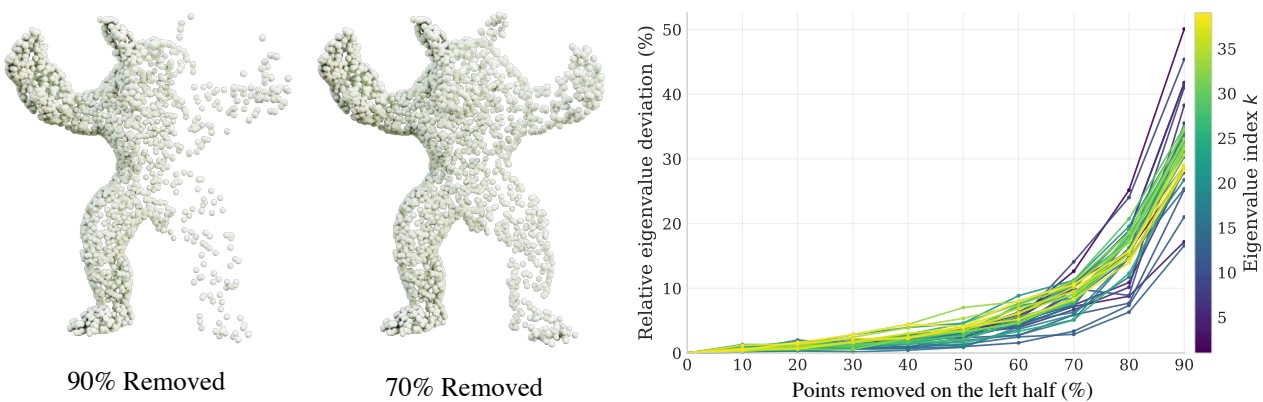

90% Removed  70% Removed

*Figure 9.* Stability of the $Q$ operator with varying sampling density. (Left) The Armadillo shape with 70% and 90% of the points removed from its left half. (Right) Relative distortion of the first 40 estimated Laplacian eigenvalues with a varying downsampling ratio. Mass is re-estimated using Kernel Density Estimation at each level. The low-frequency spectrum remains highly stable even under severe nonuniform degradation.



*Figure 10.* Functional maps computed between the first 40 eigenvectors of the normalized diffusion operator at the baseline scale $\sigma_0$ and perturbed scales ($\sigma = \alpha\sigma_0$). The near-diagonal structure indicates that the low-frequency eigenspaces are stable under this amount of noise.

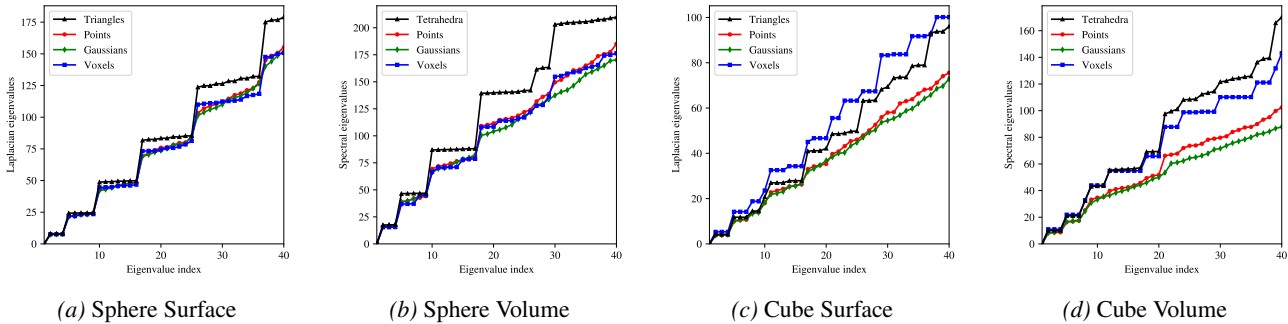

*(a)* Sphere Surface     *(b)* Sphere Volume     *(c)* Cube Surface     *(d)* Cube Volume

*Figure 11.* Laplacian eigenvalues for the sphere of diameter 1 and the cube of edge length 1.

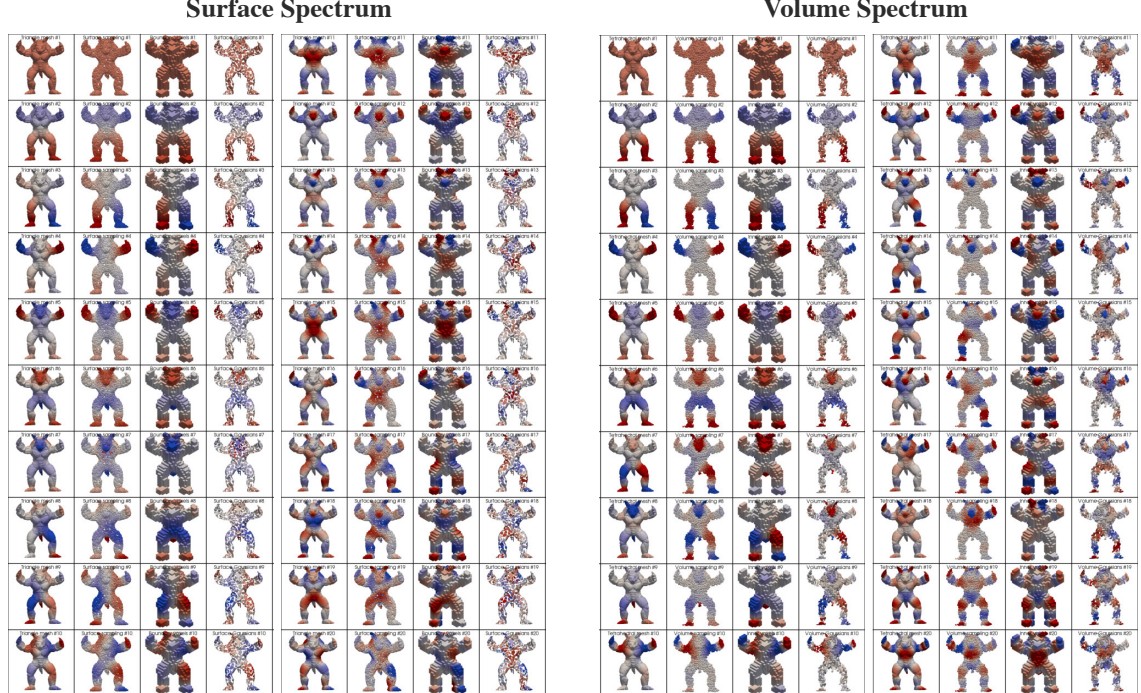

*Figure 12.* First 20 eigenvectors on a the armadillo shape treated as a surface on the left and as a volume on the right. For both volume and surfaces, the shape is encoded as simplex, a point cloud sampled uniformly at random, a voxel grid and a Gaussian mixture.

**Surface Spectrum**  **Volume Spectrum**

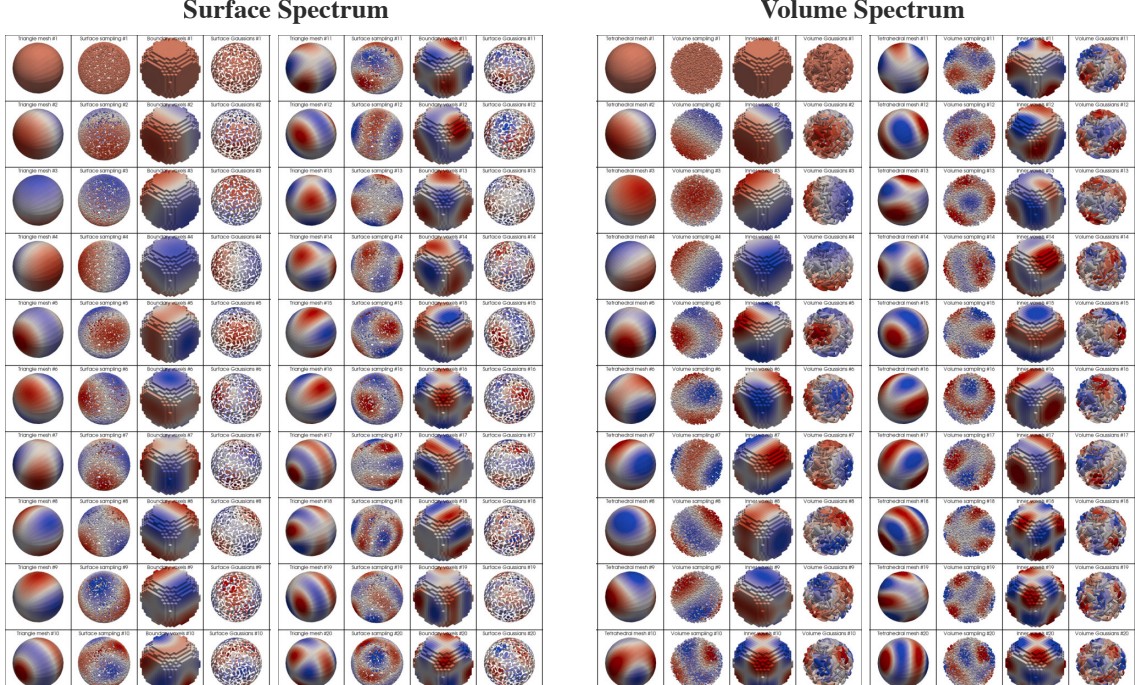

*Figure 13.* First 20 eigenvectors on a cube with edge length 1, treated as a surface on the left and as a volume on the right. For both volume and surfaces, the shape is encoded as simplex, a point cloud sampled uniformly at random, a voxel grid and a Gaussian mixture.

**Surface Spectrum**  **Volume Spectrum**

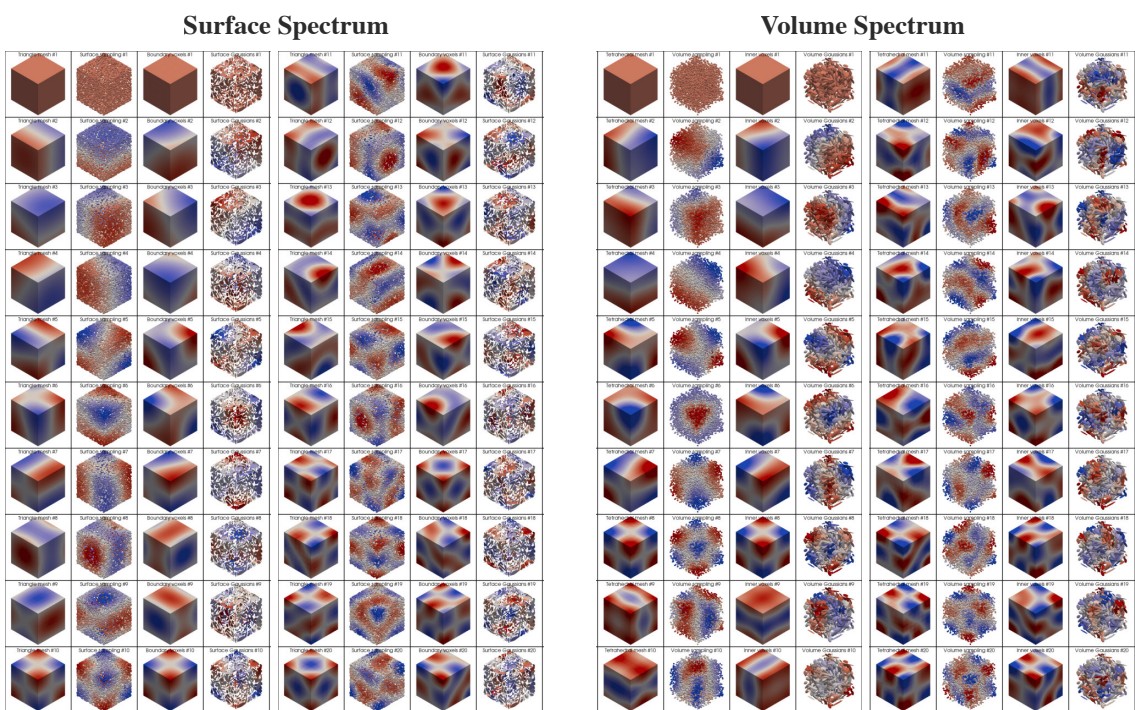

*Figure 14.* First 20 eigenvectors on a cube with edge length 1, treated as a surface on the left and as a volume on the right. For both volume and surfaces, the shape is encoded as simplex, a point cloud sampled uniformly at random, a voxel grid and a Gaussian mixture.

where $d$ is equal to 2 for surfaces and 3 for volumes. This formula is easy to compute and introduces an additional factor, the average trace of the covariance matrices $\Sigma_i$. For volumes, it relies on the observation that when all covariance matrices are equal to a constant isotropic matrix $\Sigma = \tau^2 I_3$ with trace $3\tau^2$, the smoothing operator defined in Eq. (5) of the main paper is equivalent to a Gaussian kernel matrix of variance $\sigma^2 + 2\tau^2 = \sigma^2 + (2/3)\operatorname{trace}(\Sigma)$.

Likewise, for surfaces, we expect that a regular sampling will lead to covariance matrices that have one zero eigenvalue (in the normal direction) and two non-zero eigenvalues (in the tangent plane), typically equal to a constant $\tau^2$. This leads to the formula $\sigma^2 + 2\tau^2 = \sigma^2 + (2/2)\operatorname{trace}(\Sigma)$.

**Spectrum on Animal Shape.** Similarly to Figure 3, we display on Figure 8 the 8 th eigenvector for a galloping horse shape from the Sumner dataset (Sumner & Popović, 2004), using different representation modalities.

**Stability to Sampling Density.** In Figure 9, we analyze the stability of our $Q$ operator to non-uniform sampling densities. We downsample the left half of the armadillo shape and check the absolute relative eigenvalue deviation. At each downsampling level, the mass matrix is re-estimated with kernel density estimation. We note that the spectrum of $Q$ remains particularly stable with up to $70\%$ of the samples missing on the left half.

**Stability to Bandwith Parameter.** We also study the behavior of the spectrum of $Q$ when varying $\sigma$ by a multiplicative factor $\alpha$ around the baseline scale $\sigma_0$ on the Armadillo point cloud. Since individual eigenvalues and eigenvectors are not directly comparable when the scales $\sigma$ are very different, we instead visualize in Figure 10 the functional map between the spectrum of $Q$ computed with $\sigma = \sigma_0$ and the one computed with $\sigma = \alpha\sigma_0$. The near diagonal structure shows that the eigenspace are comparable. However, this structure seems to degrade outside of the range $\alpha \in [0.5, 2]$.

**Estimated Spectrum on Standard Shapes.** In Figure 12, we display the first 20 eigenvectors of our operators defined on the Armadillo, as a complement to Figure 3 in the main paper. As expected, these mostly coincide with each other.

Going further, we perform the exact same experiment with a sphere of diameter 1 in Figures 13, as well as a cube with edge length 1 in Figures 14. The relevant spectra are displayed in Figure 11. We recover the expected symmetries, which correspond to the plateaus in the spectra and the fact that the eigenvectors cannot be directly identified with each other. We deliberately choose coarse point cloud and Gaussian mixture representations, which allow us to test the robustness of our approach. Although the Laplacian eigenvalues tend to have a slower growth on noisy data, the eigenvectors remain qualitatively relevant.

## H. Details on Gradient Flow

**The Energy Distance.** Inspired by the theoretical literature on sampling and gradient flows, we perform a simple gradient descent experiment on the *Energy Distance* between two empirical distributions. Given a source (prior) distribution $\mu = \frac{1}{N}\sum_{i=1}^{N}\delta_{x_i}$ and a target distribution $\nu = \frac{1}{M}\sum_{j=1}^{M}\delta_{y_j}$, this loss function is defined as:

$$E(\mu, \nu) = \frac{1}{NM}\sum_{i}^{N}\sum_{j}^{M}\|x_i - y_j\| - \frac{1}{2N^2}\sum_{i,j=1}^{N}\|x_i - x_j\| - \frac{1}{2M^2}\sum_{i,j=1}^{M}\|y_i - y_j\| \tag{52}$$

When all points are distinct from each other, its gradient with respect to the positions of the source samples is:

$$\nabla_{x_i}E(\mu, \nu) = \frac{1}{NM}\sum_{j=1}^{M}\frac{y_j - x_i}{\|y_j - x_i\|} - \frac{1}{N^2}\sum_{j\neq i}\frac{x_j - x_i}{\|x_j - x_i\|} \tag{53}$$

We implement this formula efficiently using the KeOps library.

**Particle Flow.** Starting from point positions $x_i^{(0)}$, we then update the point positions iteratively using:

$$x_i^{(t+\eta)} \leftarrow x_i^{(t)} - \eta NL\nabla_{x_i^{(t)}}E\left(\frac{1}{N}\sum_{i=1}^{N}\delta_{x_i^{(t)}}, \nu\right), \tag{54}$$

where $L$ is an arbitrary linear operator and $\eta$ is a positive step size. When $L$ is the identity, this scheme corresponds to an explicit Euler integration of the Wasserstein gradient flow: Figure 15 is equivalent to classical simulations such as the first row of Figure 5 in Feydy et al. (2019).

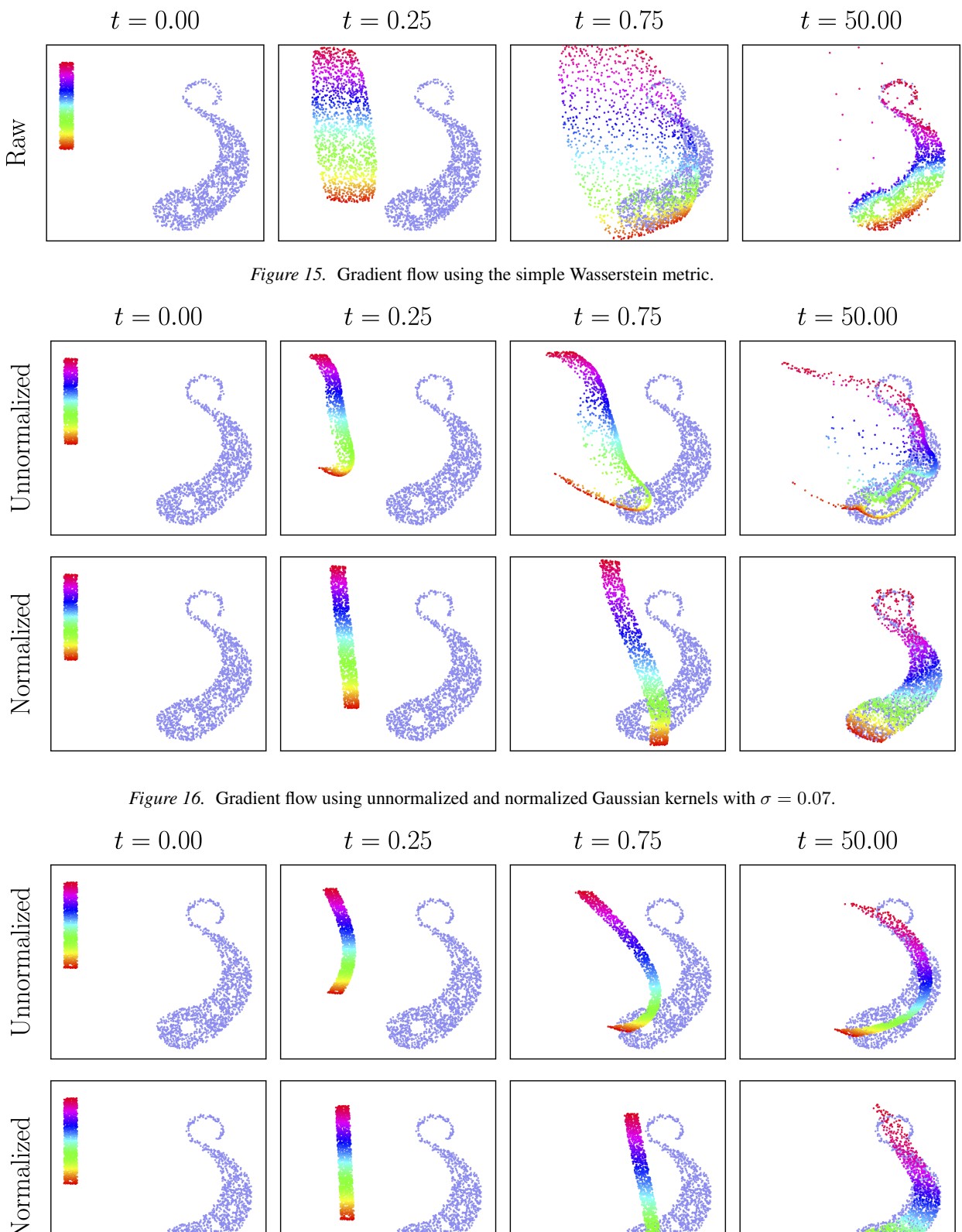

*Figure 15.* Gradient flow using the simple Wasserstein metric.

*Figure 16.* Gradient flow using unnormalized and normalized Gaussian kernels with $\sigma = 0.07$.

*Figure 17.* Gradient flow using unnormalized and normalized Gaussian kernels with $\sigma = 0.2$.

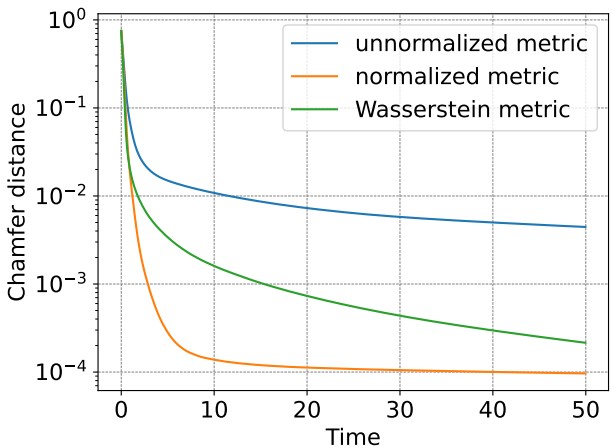

*Figure 18*
Chamfer distance between input and target distribution over time.

**Setup.** Going further, we study the impact of different smoothing operators $L$ that act as regularizers on the displacement field. We consider both unnormalized Gaussian kernel matrices and their normalized counterparts as choices for $L$, with standard deviation $\sigma = 0.07$ in Figure 4 of the main paper and Figure 16, as well as $\sigma = 0.2$ in a secondary experiment showcased in Figure 17. For each case, we run $T = 1\,000$ iterations with a step size $\eta = 0.05$.

Out of the box, unnormalized kernel matrices tend to aggregate many points and thus inflate gradients. To get comparable visualizations, we divide the unnormalized kernel matrix by the average of its row-wise sums at time $t = 0$. This corresponds to an adjustment of the learning rate, which is not required for the descents with respect to the Wasserstein metric or with our normalized diffusion operators.

As a source distribution, we use a uniform sampling ($N = 1\,500$) of a small rectangle in the unit square $[0, 1]^2$. The target distribution is also sampled with $M = 1\,500$ points using a reference image provided by the Geomloss library (Feydy et al., 2019). The entire optimization process takes a few seconds on a GeForce RTX 3060 Mobile GPU using KeOps for kernel computation.

**Visualization.** In Figure 15, we show a baseline gradient descent for the Wasserstein metric ($L = I$). As expected, the gradient flow heavily deforms the source distribution and leaves "stragglers" behind due to the vanishing gradient of the Energy Distance.

In Figure 16, we compare both kernel variants with $\sigma = 0.07$ as in the main paper. In Figure 17, we use a larger standard deviation $\sigma = 0.2$. In this case, the unregularized flow is more stable but still tends to overly contract the shape. In contrast, our normalized kernel consistently preserves the structural integrity of the source distribution.

**Quantitative Metric.** We also report in Figure 18 the Chamfer distance between the input and target distribution over time, with $\sigma = 0.07$ for unnormalized and normalized Gaussian kernels. On this Figure, we see that normalizing the kernel provides faster convergence and leads to a better optimum than standard or naively smoothed gradient.

## I. Details on Normalized Shape Metrics

**Hamiltonian Geodesic Shooting.** To compute our shape geodesics, we implement the LDDMM framework on point clouds as done by the Deformetrica software (Bône et al., 2018). If $(x_1, \ldots, x_N)$ denotes the set of vertices of the source mesh in 3D, we define the standard Hamiltonian $H(q, p)$ on (position, momentum) pairs $\mathbb{R}^{N \times 3} \times \mathbb{R}^{N \times 3}$ with:

$$H(q, p) = \tfrac{1}{2}\mathrm{trace}(p^\top K_q p) = \tfrac{1}{2} \sum_{i,j=1}^{N} (K_q)_{ij} \cdot p_i^\top p_j \,, \tag{55}$$

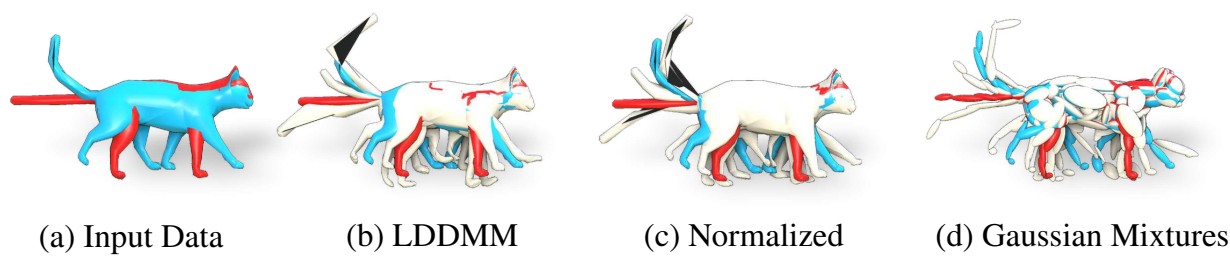

Average Area Distortion Over Time

*Figure 19.* Average face area distortion along the geodesic path, described by the time parameter. The reference area is the one of the source shape.

| (a) Input Data | (b) LDDMM | (c) Normalized | (d) Gaussian Mixtures |

*Figure 20.* **Pose interpolation and extrapolation of a cat mesh.** (a) We interpolate between source ($t = 0$, red) and target ($t = 1$, blue) poses, and extrapolate to $t = -0.5, 0.5$, and $1.5$. Note the defects in the target mesh. (b) The standard LDDMM geodesic with a Gaussian kernel ($\sigma = 0.1$) produces unrealistic extrapolations. (c) Using our normalized diffusion yields a smoother, more plausible path. (d) The method remains robust on coarse Gaussian-mixture inputs.

where $(K_q)_{ij} = \exp(-\|q_i - q_j\|^2/2\sigma^2)$ is a Gaussian kernel matrix. Starting from a source position $q(t = 0) = (x_1, \ldots, x_N)$, geodesic shape trajectories are parametrized by an initial momentum $p(t = 0)$ and flow along the coupled geodesic equation in phase space:

$$\dot{q}(t) = +\frac{\partial H}{\partial p}(q(t), p(t)), \qquad \dot{p}(t) = -\frac{\partial H}{\partial q}(q(t), p(t)) \qquad (56)$$

that we integrate to time $t = 1$ using an explicit Euler scheme with a step size $\delta t = 0.1$. The partial derivatives of the Hamiltonian are computed automatically with PyTorch.

**Shape Interpolation.** In Figure 5a, we display the source position $q(t = 0) = (x_1, \ldots, x_N)$ in red and a target configuration in blue. In Figure 5b, we use the L-BFGS algorithm to optimize with respect to the initial shooting momentum $p(t = 0)$ the mean squared error between the position of the geodesic $q(t = 1)$ at time $t = 1$ and the target configuration. Then, we use Eq. (56) to sample the geodesic curve $q(t)$ at time $t = -0.5, t = 0.5$ and $t = 1.5$.

In Figure 5c, we use the exact same implementation but normalize the Gaussian kernel matrix $K_q$ into a diffusion operator $Q_q$ before defining a "normalized" Hamiltonian $H(q, p)$:

$$H(q, p) = \tfrac{1}{2}\text{trace}(p^\top Q_q p) . \qquad (57)$$

For the sake of simplicity, we do not use the mesh connectivity information and rely instead on constant weights to define the mass matrix $M$. Although our Hamiltonian is now defined via the iterative Sinkhorn algorithm, automatic differentiation lets us perform geodesic shooting without problems. Finally, in Figure 5d, we identify every Gaussian component $\mathcal{N}(x_i, \Sigma_i)$ with a cloud of 6 points sampled at $(x_i \pm s_{i,1}e_{i,1}, x_i \pm s_{i,2}e_{i,2}, x_i \pm s_{i,3}e_{i,3})$, where $e_{i,1}, e_{i,2}$ and $e_{i,3}$ denote the eigenvectors of $\Sigma_i$ with eigenvalues $s_{i,1}^2, s_{i,2}^2$ and $s_{i,3}^2$. This allows us to use the same underlying point cloud implementation.'

*Table 4.* Shape matching performance using aligned XYZ coordinates as input features instead of WKS. This strictly point-based setting isolates the models from any implicit mesh information.

| | Method | FAUST | SCAPE | SHREC |
|---|---|---|---|---|
| *Points* | DiffNet (PC) | 7.00 | 7.45 | 9.35 |
| | ULRSSM (PC) | 6.89 | 5.60 | 10.5 |
| | Q-DiffNet (Q-FM) | 6.91 | 6.56 | 7.68 |

**Quantitative Evaluation.**    On Figure 19, we display the evolution of area distortion over time from the hand example in Figure 5. We note that, as noted in Micheli et al. (2012), standard LDDMM framework favors contraction-expansion dynamics, where area shrinks between source and target, then explodes when extrapolating. In contrast, the normalized kernel keeps a stable area at all times, even during extrapolation.

**Additional Example.**    Figure 20 displays a similar experiment as the one presented in Figure 5, but applied to a pair of cat meshes from the Sumner dataset (Sumner & Popović, 2004). Note that the target mesh in blue has inverted triangles, which remain present in the extrapolated versions of the mesh. Focusing on the cat paws, we see again a strong regularizing effect of the kernel normalization on extrapolations.

## J. Details on Point Features Learning

**Q-DiffNet.**    DiffusionNet (Sharp et al., 2022) is a powerful baseline for learning pointwise features on meshes and point clouds. It relies on two main components: a diffusion block and a gradient feature block. The diffusion block approximates heat diffusion spectrally rather than solving a sparse linear system. In Q-DiffNet, we replace this truncated spectral diffusion with our normalized diffusion operator, using a Gaussian convolution as the original smoothing operator. Given a shape $\mathcal{S}$ with vertices $X \in \mathbb{R}^{N \times 3}$, features $f \in \mathbb{R}^{N \times P}$ and a diagonal mass matrix $M$, we define:

$$\text{Q-Diff}(f, X, M; \sigma) = Q(X, M, \sigma)^m f \quad \text{where} \quad Q(X, M, \sigma) = \Lambda_\sigma K_\sigma(X, X) M \Lambda_\sigma \ . \tag{58}$$

In the above equation, $K_\sigma$ is a Gaussian kernel of standard deviation $\sigma$, $\Lambda_\sigma$ is the diagonal scaling matrix computed from Algorithm 1, and $m$ is the number of application of the operator. The scaling $\Lambda_\sigma$ is recomputed in real time at each forward pass using 10 iteration of Algorithm 1, without backpropagation. Repeating the operator $m$ times allows for simulating longer diffusion times. In practice, we use $m = 2$.

Like DiffusionNet (Sharp et al., 2022), Q-DiffNet supports multi-scale diffusion: the layer takes input features of shape $B \times C \times N \times P$ and applies separate diffusion per channel, using learnable scales $(\sigma_c)_{c=1}^C$. In DiffusionNet, typical values are $C = 256$, $P = 1$. For speed efficiency, we use $C = 32$, $P = 8$ in our experiments, which preserves the total amount of features in the network.

**Architecture Integration.**    We integrate Q-DiffNet into the ULRSSM pipeline (Cao et al., 2023), which is designed for unsupervised 3D shape correspondence. This framework trains a single network $\mathcal{N}_\Theta$, usually DiffusionNet (Sharp et al., 2022), that outputs pointwise features for any input shape $\mathcal{S}$. Inputs to the network typically consist of spectral descriptors such as WKS (Aubry et al., 2011) or HKS (Sun et al., 2009), where WKS is generally preferred. Raw 3D coordinates $(xyz)$ are also used in some settings.

**Point Cloud Inputs.**    Although DiffusionNet (Sharp et al., 2022) can operate on point clouds since it does not require mesh connectivity, it still depends on (approximate) Laplacian eigenvectors (Sharp & Crane, 2020). When working with point clouds, spectral descriptors like WKS (Aubry et al., 2011) change because they rely on the underlying Laplacian eigendecomposition. To isolate the effect of our proposed Q-operator, we minimize variability across experiments and use WKS descriptors (Aubry et al., 2011) computed from the mesh-based Laplacian for all experiments, even when retraining DiffusionNet (Sharp et al., 2022) or ULRSSM (Cao et al., 2023) on point clouds. This ensures consistent inputs across surface and point-based variants.

**Ablation on Input Features (XYZ).**    While using WKS inputs isolates the effect of the diffusion operator, we acknowledge that it propagates implicit mesh information into the network. To evaluate the methods in a strictly point-based setting, we

design an additional experiment using aligned spatial coordinates (XYZ) as input features. As shown in Table 4, replacing WKS with raw XYZ coordinates significantly degrades performance across all methods, which shows that well engineered input features heavily drive performance in current benchmarks. In that scenario Q-DiffNet remains highly competitive and achieves the best performance on SHREC19, the most challenging dataset. This confirms that Q-DiffNet can efficiently process unstructured point cloud data.

**Loss Functions.** The ULRSSM pipeline uses Laplacian eigenvectors during training to compute functional maps $C_{12}$ and $C_{21}$ from the predicted pointwise features $f_1$ and $f_2$ (Donati et al., 2020; Sharp et al., 2022; Cao et al., 2023). These maps are fed in the original ULRSSM losses: orthogonality, bijectivity, and alignment (Cao et al., 2023). For consistency, we retain these losses unchanged. Mesh-based models use ground-truth mesh eigenvectors, point cloud versions use approximate point cloud eigenvectors. Our Q-DiffNet model uses mesh eigenvectors, while the Q-DiffNet (Q-FM) variant uses eigenvectors derived from our normalized diffusion operator.

**Dataset.** We train on the standard aggregation of the remeshed FAUST (Bogo et al., 2014; Ren et al., 2019) and SCAPE datasets (Anguelov et al., 2005), using only intra-dataset pairs within the training split. We follow the standard train/test splits used in prior baselines (Donati et al., 2020; Sharp et al., 2022; Cao et al., 2023). The 44 shapes from the remeshed SHREC dataset (Melzi et al., 2019) are reserved exclusively for evaluation.

**Training.** We use the exact ULRSSM setup (Cao et al., 2023), where we train the network for 5 epochs with a batch size of 1, using Adam optimizer with an initial learning rate of $10^{-3}$ and cosine annealing down to $10^{-4}$. Training takes 6h on a single V100 GPU.

