# OpenReview forum: "Sinkhorn Normalization of Diffusion Kernels"
_ICML.cc/2026/Conference — ICML 2026 regular_

### Official Review · Reviewer_pRQL · 2026-03-10

**Soundness:** 4
**Presentation:** 4
**Significance:** 3
**Originality:** 2
**Overall Recommendation:** 5
**Confidence:** 4

**Summary:**

This paper provides a general schematic for bi-stochastic normalization of kernel affinity matrices. The key point is that this normalization produces an operator that satisfies several properties desirable of a diffusion operator, with mass preservation being the most essential. They go on to show operator point convergence in "fixed bandwidth" settings, a setting not typically of primary interest in previous research. This also motivates the implementation of the novel Q-DiffNet architecture toward problems shuch as shape correspondence.

**Compliance With Llm Reviewing Policy:**

Affirmed.

**Final Justification:**

I maintain my positive review. My assessment remains largely the same, that the paper is well constructed with elegant theory and application, but limited novelty in a purely mathematical sense.

**Key Questions For Authors:**

1. Is it possible to provide further depth at a technical level for when bi-stochastic normalization is desirable? I understand there is a lot of high level explanation for this, and this is somewhat typical in the literature, but it would be interesting if a more concrete statistical guarantee was possible

2. Have you considered verifying any stronger notions of convergence to the identified integral operator? For instance, a spectral convergence as in works such as https://arxiv.org/abs/1910.13476. It does not seem that this is covered by your pointwise guarantee, although you do demonstrate it numerically.

3. Do you have any understanding of how this procedure interacts with the data boundary? The convergence to the integral operator remains valid, but are there any pros and cons to bi-stochastic normalization in such settings, particularly in your applications of interest? For example, see https://arxiv.org/pdf/2411.18942v2 where it was shown to be beneficial for boundary detection.

**Limitations:**

Yes

**Strengths And Weaknesses:**

This paper is very polished and fills a hole in the literature, broaching the properties and asymptotic behavior of "fixed bandwidth" bi-stochastic normalized kernels and their continuum limits. Much of the theory in this paper covers well trodden ground, as the authors note. Diffusion approximation is essentially classical, and bi-stochastic normalizations have been considered for at least a decade (much longer outside the context of diffusion of course). Thus the originality and novelty of this paper are on the weak end. Indeed, the claim that previous literature has not considered the convergence of fixed bandwidth operators is not entirely accurate. In most analyses, the argument is broken into a "bias" term (this continuum operator) and a "variance" term (the error induced by discretization). As the typical setting is for random samples, this analysis yields that the empirical operator converges at a $n^{-1/2}$ rate to the continuum. While this argument is most typical for diffusion map type estimators, variants appear for the bi-stochastic Laplacian, see Cheng and Landa for example. The stated result is a bit more general, however the ingredients to prove this are standard in the entropic optimal transport literature.

This being said, the thoroughness of the paper, and axiomitization of this procedure is fine work. The Q-DiffNet architecture is also quite appealing and natural, and alleviates the direct spectral computations necessitated by DiffusionNet

---

> ### Author Rebuttal · Authors · 2026-03-30
>
> We are very grateful for the positive review and the expertise brought by the reviewer. The references linked in the review were very valuable.
> We address the concerns and questions below:
>
>
> - The reviewer is right that the **fixed-bandwidth convergence** is related to the "variance" term of bias-variance decompositions in the manifold learning literature. We will revise the paper to acknowledge this clearly, and state that we formalize and extend the result to general discrete measures, which might not be iid samples on a manifold. We view the axiomatic framework and the bridge between manifold learning, entropic optimal transport, and geometry processing as our main contribution here.
>
>
> 1. Outside of settings where mass preservation is required, **bi-stochasticity** naturally appears whenever both an operator and its adjoint need to be well-behaved. While row-normalization is natural for general kernels, two common examples require a similar normalization of the transpose.
> The first example is differentiable programming. In that setting, a smoothing operator $K$ acts on a signal in the forward pass, but its transpose $K^\top$ appears in backpropagation, where it acts as a smoothing of gradients. Bi-stochasticity ensures controlled behavior in both cases.
> The second example is LDDMM registration, where the smoothing kernel also defines an inner product. In that case, the symmetry of the operator calls for bi-stochasticity.
> We hope this provides some technical depth on scenarios where this normalization is desirable. More generally, we acknowledge that obtaining more formal statistical guarantees would be an interesting direction.
>
>
>
> 2. We believe we should be able to follow the procedure from Von Luxburg et al. (2008) to obtain **spectral convergence** from our pointwise convergence. Unlike their operator, Sinkhorn normalization ensures our operators $Q^t$ have a controlled operator norm $||Q^t||\le1$.  We believe collectively compact convergence to the integral operator might be achievable using similar arguments as those developed in Von Luxburg et al. (2008). However, there are still some technical obstructions that would need to be overcome to obtain a complete proof.
>
>
> 3. We thank the reviewer for pointing out to this very interesting and relevant paper. The Sinkhorn potentials indeed increase near boundaries to compensate for missing information.
> A significant advantage of this normalization at boundaries can be seen in the gradient flow experiment (Figure 4), where points at the boundary are pulled back slower than interior points in the absence of normalization.
> However, the inflated coefficients near boundaries might weight such points more heavily in the diffusion process. In our experiment, we did not observe any specific bias towards boundaries, but extensive evaluation, on partial shape matching for instance, would be interesting to add.
>
>
> Von Luxburg, U., Belkin, M., & Bousquet, O. (2008). Consistency of spectral clustering. The Annals of Statistics, 555-586.

---

> > ### Author Rebuttal · Reviewer_pRQL · 2026-04-02
> >
> > Thank you to the authors. I will keep my score, as it is already high, but thank you for the interesting discussion and the ideas on spectral convergence. The suggested paper is not a resource I was familiar with!

---

### Official Review · Reviewer_BakU · 2026-03-10

**Soundness:** 3
**Presentation:** 1
**Significance:** 3
**Originality:** 2
**Overall Recommendation:** 4
**Confidence:** 3

**Summary:**

The paper proposes a method for obtaining the diffusion operator $Q$ with efficient computation for the normalization. Theorem 4.4 shows that one can obtain the diffusion operator from the smoothing operator via Algorithm 1, which is a variant of the Sinkhorn algorithm. The resulting operator can be interpreted as a normalization of the raw smoothing operator $S$. Theorem 4.5 provides the theoretical guarantee that if the operator $K$ is constructed using a Gaussian or exponential kernel, then weak convergence of the measure implies uniform convergence of the operator $Q$ to the true diffusion operator. Authors also provided several numerical experiments.

**Compliance With Llm Reviewing Policy:**

Affirmed.

**Final Justification:**

The paper introduces new diffusion operator learning algorithm which can incorporate the mass-conservation property.

The main results of the paper, Theorems 4.4 and 4.5, appear to be correct. although I should upfront that, while I checked the overall logic of the proofs, I did not verify every detailed step. These results also seem to have meaningful implications---namely, for mass-conserving diffusion operators---based on the authors' claims.

However, I would like to note that I am not an expert in geometric data processing tasks; my expertise is more closely related to the Sinkhorn algorithm. Therefore, beyond the theoretical soundness of the paper, I may not be the best reviewer to assess the novelty of these implications. I also found the paper quite difficult to read. That said, the authors stated during the rebuttal that they would improve the clarity of the writing.

In summary, given the theoretical soundness of the paper, I would recommend weak acceptance. However, given my lack of expertise in this area and the paper’s unclear writing, I would avoid expressing stronger support than weak acceptance.

**Key Questions For Authors:**

* I have addressed the points that were unclear to me in the weakness section. Could the authors clarify the issues mentioned there?

**Limitations:**

yes

**Strengths And Weaknesses:**

**Strength**

* The theoretical motivation and the results are solid, in my opinion.

* There are sound theoretical guarantees for the algorithm (Theorems 4.4 and 4.5). I think the proofs are correct.

* The numerical results demonstrate the advantages of the method for certain tasks.

**Weakness**

Most of my concerns are related to the writing and presentation of the paper.

* I think the title is misleading. It seems that the authors included the term optimal transport because the method uses the Sinkhorn algorithm and the resulting $Q$ can be interpreted as the coupling derived from the entropic OT solution (as discussed in Appendices B and C). However, optimal transport plays only a limited role in the overall paper and appears as a proof ingredient rather than a core concept. I would suggest using the Sinkhorn terminology instead of OT (e.g., “Constructing Normalized Diffusion Kernels via the Sinkhorn Algorithm”).

* Following the previous point, it would be helpful to present the original Sinkhorn algorithm explicitly in mathematical or algorithmic form, since the Sinkhorn algorithm is a key component of the proposed method.

* It would also be beneficial to include a mathematical illustration and motivating examples that clarify the main task of the paper. As far as I understand, the primary goal is to construct a normalized diffusion operator $Q$ that behaves like a discretized heat diffusion operator, which can then be used in various geometric processing tasks (e.g., smoothing). In my opinion, this message and its motivation are important but are not clearly explained in the current presentation.

* The operator $K$ represents a positive-definite matrix encoding the locational relationship between points, such as the kernel implicitly described in Theorem 4.5. However, the first appearance of $K$ seems to occur in the message-passing paragraph (Line 145 of the second column), where it is introduced as $K=(D+A)/2$, which is only a special case for graphs. This may confuse readers who are not already familiar with the literature.

* Minor issue: Appendices B and C contain incorrect theorem labels.

* The meaning of the superscript $t$ in Theorem 4.5 is confusing. Based on my understanding, Theorem 4.5 aims to state that weak convergence of the measure implies uniform convergence of the operator. This type of convergence is typically indexed by the number of samples $n$ (since the empirical measure converges to the true measure as $n \to \infty$). If the authors intend a more general index, it should be introduced explicitly, since $t$ is already used earlier as the diffusion-time parameter, which makes the notation ambiguous.

* Other than the writing and presentation issues, the theorems and proofs appear to mainly apply known results about the Sinkhorn algorithm, rather than introducing a new approach, in my opinion.

**Summary**

* I think the motivation and the theoretical results, while they may not be novel, are solid and sound, and are supported by the numerical experiments. However, in my opinion, the paper has clarity issues and requires substantial rewriting.

---

> ### Author Rebuttal · Authors · 2026-03-30
>
> We thank the reviewer for acknowledging the theoretical interest of our work. We appreciate the multiple remarks which will help improve the exposition of the paper. Below we address the specific changes we will make:
>
> - We agree that the current **title** might be misleading, as optimal transport theory is only used for the proof of Theorem 4.5. We will update the title to "Correcting Diffusion Kernels with Sinkhorn Normalization".
> - We will explicitly refer to the original **Sinkhorn algorithm**, and note the specific difference to the symmetric variant of Algorithm 1.
> - We agree that the practical **motivation** of the work can be improved by providing more context in the introduction and updating Figure 1 to showcase better practical usage. We will update those.
> - We apologize for the confusion regarding the operator $K$. We will clarify that using $K=\frac{1}{2} (D+A)$ in the message-passing section was an arbitrary choice, which was only used for a toy example.
> - We thank the reviewer for catching the typos in the theorem labels in **Appendices B and C**.
> - Regarding Theorem 4.5, we agree that the notation $\mu^t$ is confusing, and will change it to $\left(\mu^s\right)_{s\in\mathbb{N}}$ to avoid conflict with diffusion time. We intentionally avoided indexing using the number of points $n$ as our result applies to a general sequence of discrete finite measures converging weakly. In particular, we do not expect empirical measures from *iid* samples on a manifold.
> - The proofs of our theorems indeed apply known results from the entropic optimal transport literature. However, our principal contribution lies in synthesizing objects from the manifold learning and optimal transport literature into a practical framework for geometry processing. Rather than focusing on Laplacian approximation, or optimality of the transport plan, we seek to obtain coherent diffusion at fixed scale. We will emphasize this more clearly in the revised version.

---

> > ### Author Rebuttal · Reviewer_BakU · 2026-04-01
> >
> > I appreciate authors for their response. I will keep my positive score.

---

### Official Review · Reviewer_wjJn · 2026-03-12

**Soundness:** 2
**Presentation:** 2
**Significance:** 2
**Originality:** 2
**Overall Recommendation:** 3
**Confidence:** 2

**Summary:**

This paper aims to overcome the limitation of existing Laplacian-based diffusion methods, which rely solely on well-structured mesh data. The authors sought to generate a "diffusion operator" that preserves the key properties of the Laplacian — namely mass conservation and symmetry — by normalizing an arbitrary similarity/adjacency matrix through the Symmetric Sinkhorn algorithm.

**Compliance With Llm Reviewing Policy:**

Affirmed.

**Final Justification:**

While I appreciate the clarifications, the core concern remains that the empirical results do not fully support the claim of Laplacian-independence, as the method still relies on mesh-derived inputs. The rebuttal narrows the scope rather than resolving this issue. Overall, I am not convinced that the main claim is sufficiently validated.

**Key Questions For Authors:**

1. The proposed method (Sinkhorn normalization) was benchmarked running on a GPU (V100), while the baseline (Implicit Euler) was run on a CPU (Xeon Gold). Shouldn't a fair comparison also include both methods evaluated on CPU?

2. The visualized results are largely limited to shapes with strong symmetry — such as spheres and cubes — or geometrically distinctive models like the Armadillo. Does the method maintain a comparable level of fidelity on real-world scan data with significantly higher noise levels or extremely low sampling density?

3. When σ is set to a small value on complex shapes, could the Sinkhorn normalization fail to converge or become numerically unstable?

4. How should the diagonal scaling factors Λ_ii be interpreted near boundaries or in regions with sparse sampling? Since the Sinkhorn normalization enforces global stochasticity, could this introduce non-local corrections that potentially distort the intended locality of the diffusion operator, especially for isolated points or irregular sampling patterns?

5. While enforcing Q1=1 guarantees mass conservation and stability, could this normalization suppress meaningful geometric signals encoded in sampling density? In particular, how should one evaluate the trade-off between exact mass preservation and potential distortion of geometry when the mass matrix M is estimated with noise or sampling bias?

**Limitations:**

Yes

**Strengths And Weaknesses:**

### Strengths

It is immediately applicable to all discrete geometric data — whether or not connectivity information is present — including point clouds, voxel grids, Gaussian Mixture Models (GMMs), and graphs.
Rather than a simple approximation, it respects the core physical laws of heat diffusion grounded in Optimal Transport theory.

### Weaknesses

1. While Q-DiffNet operates on point clouds, the Wave Kernel Signature (WKS) descriptors used as input features are computed from mesh-based Laplacians. This means the model is not trained purely on point cloud information alone — it partially borrows geometric ground truth derived from meshes, which makes it difficult to consider the performance comparison a fair one.

2. The loss functions employed during training — such as orthogonality and alignment losses — still rely on (mesh-derived or approximated) eigenvectors. In other words, only the diffusion operator within the network has been replaced, while the overall system remains strongly tied to the classical Laplacian framework — does it not?

I am not deeply familiar with this area, so some of my questions may be misguided. I am happy to raise my final score upon reading the authors' clarifications and explanations.

---

> ### Author Rebuttal · Authors · 2026-03-30
>
> We thank the reviewer for the detailed comments, and for recognizing the applicability of our method for geometry processing. We address the questions below:
>
> ## Weaknesses
> Regarding the **shape matching experiment**, we appreciate the opportunity to clarify this setup. The main goal was to highlight the practical usability of our diffusion operator as a drop-in replacement for existing layers.
> In theory, our operator can approximate three quantities: heat diffusion (Q), Laplacian eigenvectors, and Laplacian eigenvalues. Eigenvalue approximation with heuristic described in Appendix G is the least reliable approximation here.
> To isolate the effect of the $Q$ operator itself, we actively tried to reduce the variability across all experiments. In this setting, *all methods* use mesh-based WKS as input.
> Regarding the losses, the Q-DiffNet experiment uses mesh-based eigenvectors, but the Q-FM variant uses the approximated eigenvectors, independent from the mesh.
>
> Overall, the system is still tied to the Laplacian because of the inputs, and so are the retrained "PC" baselines. Obtaining a pipeline fully agnostic to standard approximation, with state of the art performance is a strong direction for future work, and would require better eigenvalue approximation, or alternative inputs.
> While the detailed setup is described in Appendix J, we agree that the main text doesn't clearly state these important points, which can make the experiment confusing to the reader. We will update this in the revised version.
>
>
> ## Questions
>
> 1. Regarding the **runtimes benchmark**, we followed standard practice and evaluated the sparse solver on the CPU. While the Sinkhorn algorithm is highly efficient on GPU, its naive CPU version is notably slow without additional acceleration (e.g. using KNN). Below, we ran a CPU benchmark for Implicit Euler and a KNN version of Sinkhorn on a dual-socket Intel Xeon E5-2695 v4. Note that KNN graph construction can be seen as preprocessing, as it is independent from the diffusion scale, which is why its timing is presented separately:
>
> (*Times in seconds. KNN precomputation in parentheses*)
> | N    | Implicit Euler | Sinkhorn (k=25) | Sinkhorn (k=100) |
> | ---- | -------------- | --------------- |:---------------- |
> | 10K  | 0.428          | 0.010 (+0.117)  | 0.022 (+0.274)   |
> | 100K | 2.921          | 0.064 (+0.612)  | 0.338 (+1.819)   |
> | 1M   | 71.776         | 0.971 (+6.004)  | 5.199 (+18.351)  |
>
> 2. We agree that the examples presented are mostly **symmetric**. On noisy scans, we did observe coherent behavior of our normalized operators.
> To quantitatively check the influence of sampling on the results, we evaluated the relative variation of estimated Laplacian eigenvalues for several amounts of downsampling applied only to the *left part* of the armadillo. The mass matrix is re-estimated with KDE at each level.
> We observe that the variation of the first 40 eigenvalues stays $\le$ 5% when removing 40\% of the samples. Removing 60% of the half-shape, we get $\le$ 10% variation, and $\le$ 20% for 80% . This experiment will be added the revised version.
>
> 3. Regarding the **numerical stability for small $\sigma$**, note that $\sigma\to0$ can lead to failure on *standard* Sinkhorn. However the symmetric variant we use always converges (Knight et al., 2014), whatever the value of $\sigma$. At the limit $\sigma=0$, the normalized matrix simply converges to the identity.
>
> 4. Regarding the **value of the scaling factors at borders**, this is a great question, which, in the Gaussian case, was recently explored in Kohli et al. (2024), as linked by Reviewer `pRQL`. The normalizing factors increase at the boundary to compensate for the missing information. Regarding **non-locality**, a simple calculation shows that each scaling factor is bounded by $\lambda_i\leq\frac{1}{\sqrt{m_i}}$. Generally, for a Gaussian kernel and any pair of points $(x_i,x_j)$ at a distance $D$,the diffusion between the points is bounded by $Q_{ij}\leq \sqrt{\frac{m_j}{m_i}} \exp(-\frac{D^2}{2\sigma^2})$. Non local behaviors require the ratio of point masses to grow faster than the exponential distance decay. While we did not observe this in practice, this behavior could happen for very large diffusion scales on pathological samplings.
>
> 5. Regarding the **exact mass preservation** constraint, we entirely agree that strict mass preservation is not always optimal, for instance in graphs, where highly connected nodes might benefit from amplifying features. In our geometric context however, we treat irregular sampling as a nuisance rather than a feature, and the symmetric Sinkhorn normalization corrects this bias.
>
> Knight, P. A., Ruiz, D., & Uçar, B. (2014). A symmetry preserving algorithm for matrix scaling. SIAM journal on Matrix Analysis and Applications.
>
> Kohli, D., He, J., Holtz, C., Mishne, G., & Cloninger, A. (2024). Robust estimation of boundary using doubly stochastic scaling of Gaussian kernel. arXiv preprint arXiv:2411.18942.

---

> > ### Author Rebuttal · Reviewer_wjJn · 2026-04-03
> >
> > Thank you for the detailed responses and clarifications.
> >
> > After going through the paper and the rebuttal, I still find that some core issues remain unresolved.
> >
> > __Core concern: mismatch between claims and empirical validation__
> > The paper positions itself as enabling diffusion-like processing on unstructured data without relying on Laplacians. However, the strongest experimental results (e.g., Q-DiffNet) still depend on mesh-derived inputs such as WKS and eigenvectors, as clarified in the rebuttal.
> >
> > Because of this, it is difficult to determine whether the observed performance gains come from the proposed operator itself or from the retained geometric priors. In its current form, the empirical evidence does not fully validate the extent of the claimed Laplacian-independence in practical settings.
> >
> > __Scope clarification vs. resolution__
> > The rebuttal clarifies that the current pipeline is still partially tied to the classical Laplacian framework. While this improves transparency, it effectively narrows the scope of the contribution rather than resolving the original concern.
> >
> > To better support the claim, it would be important to include experiments that are fully independent of mesh-derived quantities.

---

> > > ### Author Response · Authors · 2026-04-07
> > >
> > > We thank the reviewer for their acknowledgement, and appreciate the opportunity to clarify the experimental setup.
> > >
> > > We would like to highlight that the shape matching experiment was designed as a *controlled ablation*: we only changed the diffusion operator while keeping other blocks fixed. This setup is precisely what allows to isolate the effect of the operator. Using the same WKS inputs for all methods, including baselines, ensures a fair comparison, where any performance change can only be attributed to one component.
> > >
> > > Our choice to use *mesh* WKS as input, even for point cloud baselines, came from a similar observation. Using bad input features lead to poor performance for all method, making it hard to meaningfully compare feature extractors. Providing all methods with *the same good and informative features* allows to study the general ability of the network to process features, without the quality of the input interfering. The XYZ experiment below shows exactly this, where performance drops for all methods.
> > >
> > > We believe that simultaneously changing inputs, feature extractor and losses would make it hard to draw any conclusion about the impact of each individual change.
> > >
> > > We also wish to clarify that the network only requires informative pointwise features as input, and that these can be easily changed. WKS was chosen as it is the default choice in current benchmarks. We believe that recent research leveraging foundation models could soon offer plausible alternative inputs. The losses and spectral blocks are also simply borrowed from existing pipelines, where spectral and mesh information are never seen by Q-DiffNet. What the experiment shows is that Q-DiffNet can efficiently process pointwise features on unstructured data.
> > >
> > > Building a pipeline that is fully mesh-independent *and* that obtains state of the art performance would require redesigning inputs, losses and architecture, which would be a full paper in itself. In contrast the shape matching experiment is here only part of the contributions of this paper.
> > >
> > >
> > > That said, we understand that using mesh WKS as input raises concerns, since mesh information is propagated in the network. Therefore, we ran an additional experiment using XYZ as inputs, using **aligned** versions of the datasets. Note the Q-DiffNet (Q-FM) model doesn't use *any* mesh quantities, i.e it uses our estimated eigenvalues/eigenvectors in the losses.
> > >
> > > | Method | FAUST | SCAPE | SHREC
> > > | - | -| - | -
> > > | DiffNet (PC) |   7.00 | 7.45 | 9.35
> > > | ULRSSM (PC) |   6.89 | 5.60  | 10.5
> > > | Q-DiffNet (Q-FM) | 6.91 | 6.56 | 7.68
> > >
> > >
> > > Performance degrades significantly for all methods compared to WKS inputs. This confirms that input features are major drivers for performance. Q-DiffNet still remains competitive and even obtains the best performance on SHREC, which is the most challenging dataset. We believe this still supports the value of our operator in this pointcloud setting, and will include this experiment in the revised version.
> > >
> > > We will additionally heavily clarify the setup and the goal of this experiment in the revised version.

---

### Official Review · Reviewer_3pyx · 2026-03-13

**Soundness:** 3
**Presentation:** 3
**Significance:** 3
**Originality:** 3
**Overall Recommendation:** 4
**Confidence:** 4

**Summary:**

Laplacian smoothing is a well defined operation in many domains. But in unstructured domains where it is not clear how to build the laplacian or when the laplacian is costly to build convolution kernels and message passing are often used for smoothing. But these are biased against the boundaries of the domain. This paper attempts to solve this bias by introducing a broad class of smoothing operators, derived from general similarity or adjacency matrices, and demonstrate that they can be normalized into diffusion-like operators that inherit desirable properties from Laplacians.

The paper's approach relies on a symmetric variant of the Sinkhorn algorithm, which rescales positive smoothing operators to match the structural behavior of heat diffusion. They argue that this construction enables Laplacian-like smoothing and processing of irregular data such as point clouds, sparse voxel grids or mixture of Gaussians. Additionally, they attempt to show that the resulting operators not only approximate heat diffusion but also retain spectral information from the Laplacian itself, with applications to shape analysis and matching

**Compliance With Llm Reviewing Policy:**

Affirmed.

**Final Justification:**

The rebuttal addressed our main concerns, so we decide to keep the scores.

**Key Questions For Authors:**

see weaknesses

**Limitations:**

see weaknesses

**Strengths And Weaknesses:**

# Strengths
- Clear and easy to follow
- Really important problem tackled
# Weaknesses
- Figure 1 is unclear for the laplacian like eigen vectors, which vector is simulated the first, second, third or ....?
- The paper relies on similarities between points which are not very easy to get or decide which to use. Can the authors provide performance analysis on different potential metrics on every shape? e.g., use the metrics to get S on pointclouds also on the gaussian mixture and voxels and vice versa and motivate your choices
- what is the unit for the reported wall times in table 1?
-

---

> ### Author Rebuttal · Authors · 2026-03-30
>
> We thank the reviewer for the positive comments on the exposition, and for recognizing the importance of the problem. We address your comments below:
>
> - We apologize for the ambiguity of the teaser Figure. The eigenvectors displayed are the 2nd, 6th and 7th. We will update the Figure and its caption in the revised version to provide better intuition on the use cases of the method.
>
>
> - In all our experiments, we advocate using a Gaussian kernel for point-cloud or voxel data, and its extension to Gaussian distributions when using Gaussian mixtures.
> While exponential or other handcrafted kernels are also viable, the choice remains mostly application dependant. Theoretical guarantees show that a Gaussian kernel is a great fit for Laplace-Beltrami approximation. However for pure diffusion tasks, designing or learning other types of kernels might be beneficial. We believe this is a strong direction for future works.
> When using an isotropic Gaussian kernel, the variance of the Gaussian is a free parameter, that is set or optimized according to the scale of the data.
> For the gradient flow experiment, we observe that our normalization provides stable behaviors for several value of this parameter (see Appendix H). For spectral analysis, low frequency modes remain stable under changes of bandwidth, and will provide an illustrative experiment in the revised version.
>
>
> - The wall-clock runtimes reported in Table 1 are measured in milliseconds. We apologize for the missing unit and will update the table in the final version.

---

> > ### Author Rebuttal · Reviewer_3pyx · 2026-04-04
> >
> > - While some experiments could not be provided immediately given the rebutal format, the authors promised to add some in the revision e.g., "For spectral analysis, low frequency modes remain stable under changes of bandwidth, and will provide an illustrative experiment in the revised version".
> >
> > - After reading from Reviewer wjJn, I also wanted to ask a clarification on the experiments. Please can experiments be provided that don't use any Laplacian or mesh priors (HKS, WKS, GPS, and the likes) for the diffussion process?

---

> > > ### Author Response · Authors · 2026-04-07
> > >
> > > We thank the reviewer for their comment, and appreciate the opportunity to discuss the shape matching experiment.
> > >
> > >
> > > We would like to clarify a possible misunderstanding about this experiment. In this experiment, Q-DiffNet is a feature extractor network, that takes pointwise features as inputs and outputs new pointwise features. The diffusion process there doesn't use **any** mesh prior, be it triangles or spectral values. The dependency highlighted by reviewer `wjJn` concerned how the *input features* were obtained.
> > >
> > > The choice for using mesh-WKS as inputs for all methods, *including pointcloud baselines*, was motivated by the need for a controlled comparison. Using bad input features lead to poor performance for all methods, making it hard to meaningfully compare feature extractors. This way, we isolated the effect of the feature extractor, and therefore of our $Q$ operator. Mesh-WKS was selected as it is the default informative descriptor in current benchmarks. The network is not particularly adapted to these features, and these could be replaced by mesh-independent alternative in the future (e.g. current research studies using foundation models). We refer the reviewer to our response to Reviewer `wjJn` for a more detailed discussion of the experimental setup and the role of input features.
> > >
> > >
> > > That said, we understand that using mesh WKS as input raises concerns, since mesh information is propagated in the network. Therefore, we ran an additional experiment using XYZ as inputs, using **aligned** versions of the datasets. Note the Q-DiffNet (Q-FM) pipeline doesn't use *any* mesh quantities, i.e it uses our estimated eigenvalues/eigenvectors in the losses.
> > >
> > > | Method | FAUST | SCAPE | SHREC
> > > | - | -| - | -
> > > | DiffNet (PC) |   7.00 | 7.45 | 9.35
> > > | ULRSSM (PC) |   6.89 | 5.60  | 10.5
> > > | Q-DiffNet (Q-FM) | 6.91 | 6.56 | 7.68
> > >
> > >
> > > Performance degrades significantly for all methods compared to WKS inputs. This confirms that input features are major drivers for performance. Q-DiffNet still remains competitive and even obtains the best performance on SHREC, which is the most challenging dataset. We believe this still supports the value of our operator in this pointcloud setting, and will include this experiment in the revised version. The entire setup and goal of this experiment will also be clarified.

---

### Decision · Program_Chairs · 2026-04-30

**Decision:**

Accept (regular)

**Comment:**

The authors propose a framework for constructing diffusion-like operators from arbitrary similarity matrices via a symmetric Sinkhorn normalization. The goal is to enable Laplacian-style smoothing and dealing with irregular data such as point clouds, voxel grids, and Gaussian mixtures, where Laplace operators are not available. The proposed framework is theoretically grounded and supported by analysis showing convergence to diffusion operators. Despite the limitations raised by the reviewers on clarity, presentation, and its moderate novelty, we think the contributions are technically solid, the authors address an important problem, its finding results are interesting for the community. Therefore, we recommend acceptance.